# Evidence against tetrapod-wide digit identities and for a limited frame shift in bird wings

Thomas A. Stewart[1,2,7], Cong Liang [3,8], Justin L. Cotney [4], James P. Noonan[5], Thomas J. Sanger[6] & Günter P. Wagner [1,3]

In crown group tetrapods, individual digits are homologized in relation to a pentadactyl ground plan. However, testing hypotheses of digit homology is challenging because it is unclear whether digits represent distinct and conserved gene regulatory states. Here we show dramatic evolutionary dynamism in the gene expression profiles of digits, challenging the notion that five digits have conserved developmental identities across amniotes. Transcriptomics shows diversity in the patterns of gene expression differentiation of digits, although the anterior-most digit of the pentadactyl limb has a unique, conserved expression profile. Further, we identify a core set of transcription factors that are differentially expressed among the digits of amniote limbs; their spatial expression domains, however, vary between species. In light of these results, we reevaluate the frame shift hypothesis of avian wing evolution and conclude only the identity of the anterior-most digit has shifted position, suggesting a 1,3,4 digit identity in the bird wing.

[1] Department of Ecology and Evolutionary Biology, Yale University, New Haven, CT 06520, USA. [2] Minnesota Center for Philosophy of Science, University of Minnesota, Minneapolis, MN 55455, USA. [3] Systems Biology Institute, Yale University, West Haven, CT 06516, USA. [4] Department of Genetics and Genome Sciences, UConn Health, Farmington, CT 06030, USA. [5] Department of Genetics, Yale School of Medicine, New Haven, CT 06520, USA. [6] Department of Biology, Loyola University in Chicago, Chicago, IL 60660, USA. [7]Present address: Department of Organismal Biology and Anatomy, The University of Chicago, Chicago, IL 60637, USA. [8]Present address: Center for Applied Mathematics, Tianjin University, Tianjin 300072, China. Correspondence and requests for materials should be addressed to T.A.S. (email: tomstewart@uchicago.edu) or to G.P.W. (email: gunter.wagner@yale.edu)

Limbs evolved from paired fins in the Late Devonian, and early tetrapods possessed more than five digits on the fore- and hindlimbs[1,2]. Later in the tetrapod stem, a pentadactyl pattern stabilized as the ground plan for the limb. Individual digits are homologized between species and between fore- and hindlimbs in reference to this pentadactyl ground plan[3]. However, it remains controversial whether such hypotheses of identity correspond to distinct developmental programs among the digits (developmental identities), or just the relative position of digits along the limb's anteroposterior axis (positional identities)[4–7]. Below we use the symbols D1, D2, etc. to indicate positional identities from anterior (thumb) to posterior in the pentadactyl ground plan, rather than to indicate developmental identities.

The idea that digit homology corresponds to developmental identity is based upon several assumptions. First, that a digit's morphology reflects the expression of effector genes that control cell proliferation, differentiation, and matrix deposition (i.e., the genes that realize the digit during development). Second, that the expression of these effector genes is caused by the differential expression of genes controlling cell–cell signaling and gene transcription (i.e., signaling and transcription factor genes). Therefore, based on the observation of stable, digit-specific morphologies in development and evolution, it is predicted that individual digits will have distinct transcription factor and signaling gene expression profiles. Studies aiming to test for homology of developmental identity predict that the identities were present in a common ancestor and are conserved, detectable through comparative study of transcription factor and signaling gene expression profiles, in descendent lineages.

The anterior-most digit (D1) (e.g., human thumb) appears to have a distinct developmental identity in amniotes as compared with the more posterior digits (D2–D5). D1 is marked by a unique gene expression profile—low expression of *HoxD11* and *HoxD12* and high expression of *Zic3* relative to other digits[7–9]—and it appears able to develop independently of *Shh* signaling[9–11]. In addition, analysis of morphological variation in primates identified a high degree of variational independence of D1 relative to the more posterior digits[12]. Models of posterior digit identity have been proposed according to the relative exposure of limb bud mesenchymal cells to *Shh*, which emanates from the zone of polarizing activity prior to digit condensation[10,11]. However, broadly conserved marker genes for individual posterior digits have not been identified in the interdigital mesenchyme, the signaling center that patterns digits[13,14]. For instance, while the combinatorial expression of *Tbx2* and *Tbx3* is necessary to generate the phenotypes of D3 and D4 in chicken hindlimb[15], it is questionable whether these developmental identities are conserved in other species, like mouse, with limited morphological differentiation of the posterior digits.

Debates of digit homology are especially challenging to resolve when limbs have fewer than five digits. This problem has been most actively investigated in the tridactyl avian wing, because of the appearance of a conflict between paleontological and developmental data[16]. The fossil record of theropod dinosaurs shows a clear pattern of reduction of the posterior two digits in the lineage leading to birds, yet digits in the wing have been described as developing in the middle three positions of a pentadactyl developmental ground plan[17–22]. To explain this discrepancy, the frame shift hypothesis was proposed[16]. It posited that a homeotic shift occurred in the avian stem such that the developmental programs that were once expressed in D1, D2, and D3 are now executed in the digits that develop in positions D2, D3, and D4, respectively. Comparative analyses of gene expression have found support for this hypothesis: in situ hybridization and transcriptomics have revealed similarity between the anterior digit of the adult avian wing, which develops in position D2, and D1 of

other limbs[7,23], and cells in the zone of polarizing activity do not contribute to the skeletal tissues of the digits of the adult avian wing, a pattern consistent with digit D1–D3 of other limbs[24,25].

Analyses of developmental identity of digits tend to focus on either *Shh* expression and signaling in the limb bud prior to digit condensation or gene expression in the interdigital mesenchyme after digit condensation. In this paper, we follow the latter approach. Although *Shh* signaling determines the number of digits formed and initiates the development of differences between digits, the execution of digit-specific developmental programs continues long after the *Shh* signal has ceased. Dahn and Fallon[13] demonstrated in the chicken hindlimb that genes expressed in the interdigital mesenchyme regulate digit-specific morphologies, including the number of phalanges. Subsequent work in the chicken hindlimb showed that this signaling, in the phalanx-forming region, is active between stages 27 and 30[14]. In addition, we follow the latter approach because we allow the possibility that in digits developmental identity can be decoupled from digit position.

Here we present comparative transcriptomic data, analyzing developing digits, and their associated posterior interdigital mesenchyme in five species to test hypotheses of digit identity in amniotes. We report a surprising diversity of regulatory gene expression profiles of digits between species. Analyses further reveal a core set of transcription factor genes differentially expressed among digits and suggest an alternative model for the evolution of the bird wing.

## Results

**Disparity in digit expression profiles.** To characterize the gene expression profiles of digits in pentadactyl amniote limbs, we sequenced RNA of developing digits and their associated posterior interdigital mesenchyme from the forelimbs of mouse, green anole (*Anolis*), and American alligator (Fig. 1a). In each of these species, hierarchical cluster analysis (HCA) and principal component analysis (PCA) of the transcriptomes shows a weak signal of sample clustering by digit (Supplementary Fig. 1). The strongest signals of digit-specific expression profiles are observed in D1 of mouse and D4 of the alligator. Groupings of the other digit samples are not well supported. We hypothesized that this result might imply that any signal of gene expression differentiation among digits is overwhelmed by noise when all genes are considered, because most genes are likely irrelevant to the developmental identity of digits. If such a signal exists, we predict that it will be reflected preferentially in the expression of transcription factor and signaling genes. Therefore, we again performed HCA and PCA on the samples of each species, this time using two gene lists: a curated set of known limb patterning genes that are sensitive to *Shh* signaling ($N = 159$)[26], and transcription factor genes ($N = 2183$)[27].

In mouse and alligator, HCA and PCA of known limb patterning genes result in clustering of samples by digit (Fig. 1b, c). In mouse, D1 is strongly differentiated from the other digits. In alligator, an anterior cluster, comprised of digits D1, D2, and D3, is differentiated from a posterior cluster, comprised of D4 and D5. By contrast, analysis of known limb patterning genes in *Anolis* shows weak clustering of samples by digits (Fig. 1d). This suggests a level of homogeneity among *Anolis* digits that is not observed in either mouse or alligator. Analysis of all transcription factors for these species yields comparable results to what is recovered for limb patterning genes, but with generally lower adjusted uncertainty values in HCAs (Supplementary Fig. 2).

To further test the hypothesis that there is limited gene expression differentiation among *Anolis* digits as compared with the other pentadactyl limbs sampled, we took advantage of a

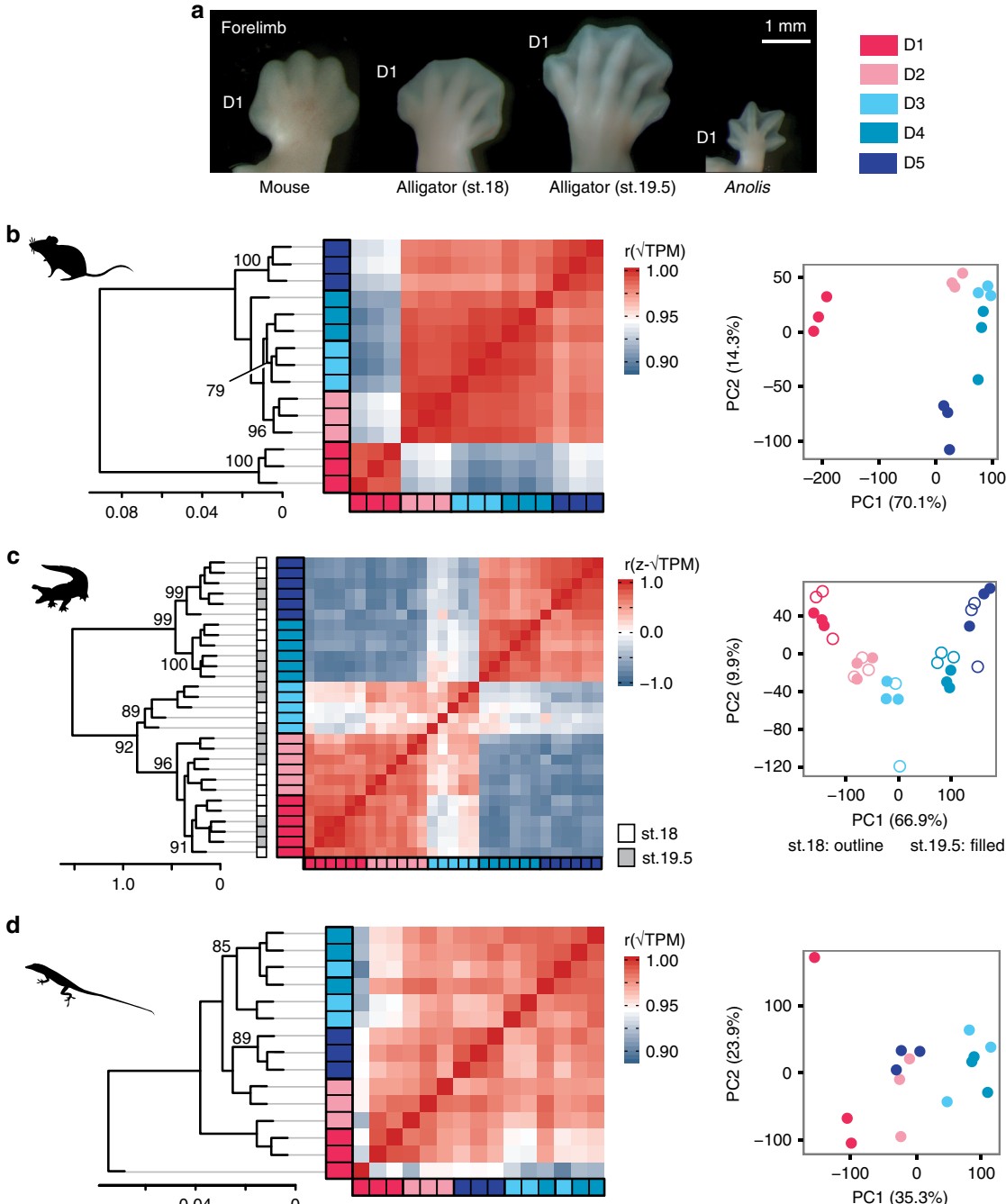

**Fig. 1** Pentadactyl amniote limbs have disparate patterns of genetic differentiation of digits. **a** Photographs of right forelimbs at the stages sampled, dorsal perspective. Analyses of limb patterning genes show that in **b** mouse and **c** alligator, replicates of each digit form clusters, indicating that the digits have distinct gene expression profiles. By contrast, **d** *Anolis* digits do not show clear differentiation of gene expression profiles. Alligator illustration reproduced with permission by Michael Richardson. *Anolis* illustration by Sarah Werning without modification (license [https://creativecommons.org/licenses/by/3.0/])

result from multiple testing theory:[28] If a differential expression analysis is conducted on two sample types that are not genetically differentiated, then the resultant frequency distribution of *p* values will be uniform within the [0, 1] interval. On the other hand, if there are truly differentially expressed genes among the compared sample types, then the *p* value distribution is expected to be biased toward *p* = 0. We conducted differential expression analyses of adjacent digits of the forelimbs of mouse, alligator, and *Anolis* using EdgeR[29,30] and inspected *p* value distributions (Fig. 2). In *Anolis*, all comparisons of adjacent digits result in

*p* value distributions that are close to uniform, suggesting that there is very weak, if any, genetic differentiation of adjacent fingers. We note that this result is independent of any *p* value significance threshold or false discovery correction method. By contrast, most of adjacent pairwise digit comparisons for mouse and alligator show a strongly biased *p* value distribution, the exception being D2 and D3 in mouse. This is consistent with the idea that, in general, most digits in a limb are genetically differentiated, while in *Anolis* genetic differentiation of digits is minimal or absent.

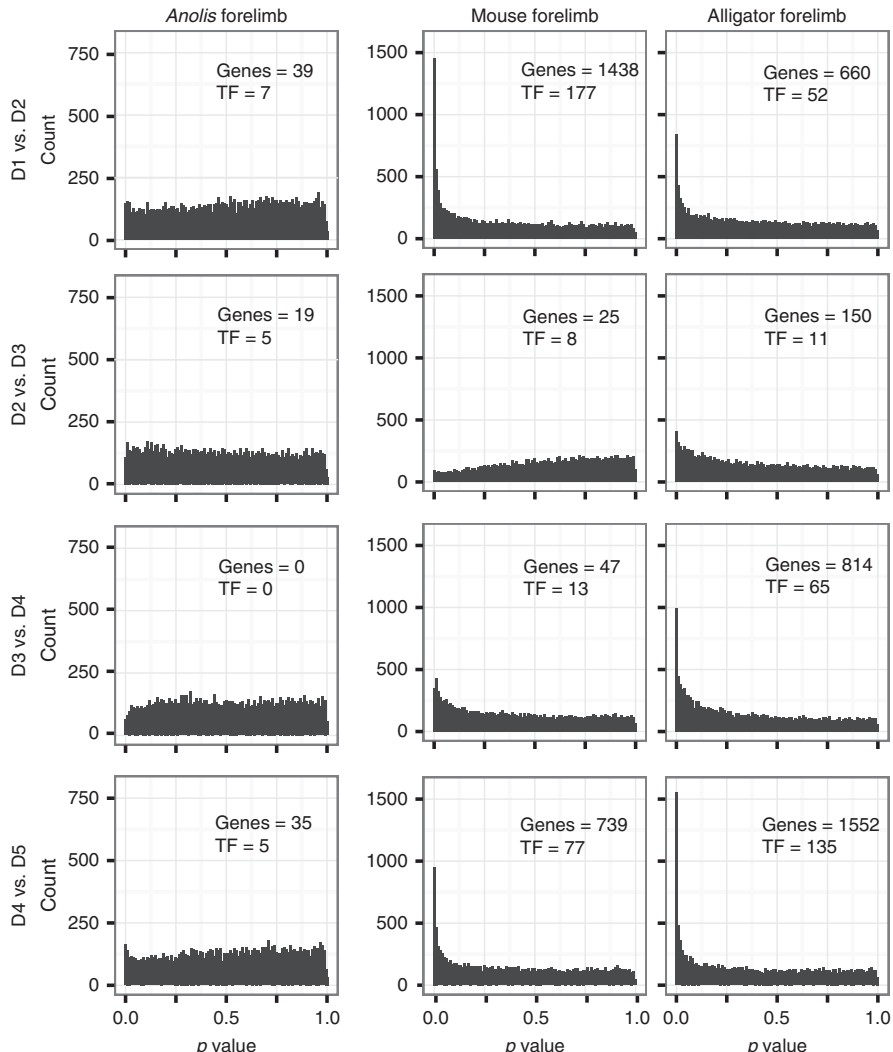

**Fig. 2** Differential expression analyses suggest homogeneity among *Anolis* digits. Histograms show the distribution of *p* values from differential expression analyses of adjacent digits. In *Anolis*, *p* value distributions that are close to uniform, indicating very weak genetic differentiation of adjacent fingers. Mouse and alligator, on the other hand, generally show strongly biased *p* value distributions. The number of genes that are identified as differentially expressed at an FDR threshold of 0.05 are noted in each panel as "genes," and the number of transcription factors among these are noted in each panel as "TF."

Given that these three limbs differ in their broad patterns of gene expression differentiation of digits, we next asked whether individual genes show divergent or constrained expression patterns across the forelimb in the different species. Specifically, we compared adjacent digits, identified differentially expressed transcription factor genes, and then assessed which differences are shared among mouse, alligator, and *Anolis*. Of the 1133 transcription factor genes that are one-to-one orthologs in these three species, only four genes are differentially expressed in a conserved pattern among corresponding adjacent digits (Fig. 3). There are three genes that differentiate D1 from D2 (*Hoxd11, Hoxd12,* and *Sall1)*, and there is one that differentiates D4 from D5 (*Tbx15*) in all three species. No transcription factors are differentially expressed in all three species between the median digits (i.e., differentiating D2 from D3, or D3 from D4).

If the homogeneity observed among *Anolis* forelimb digits is a derived condition, then this could limit our ability to diagnose plesiomorphic developmental identities. Therefore, we also considered the chicken hindlimb, which has digits D1–D4. We reanalyzed published transcriptomic data for hindlimb digits[7], mapping reads to a new chicken genome (Galgal5.0)[31]. HCA and PCA of digits of the chicken hindlimb show a unique pattern of

similarity as compared with pentadactyl limbs: an anterior cluster, comprised of D1 and D2, is differentiated from the posterior cluster, comprised of D3 and D4 (Supplementary Fig. 3). Similar to alligator, this pattern of correspondence among the digits is stable across the developmental window sampled (st. 28–31). As before, we tested for differential expression in adjacent digits and identified one-to-one orthologous transcription factor genes that are differentially expressed at the same position between mouse and alligator forelimb and chicken hindlimb (Fig. 4a, Supplementary Fig. 4). Of the 1049 transcription factor genes, ten differentiate D1 and D2 ($n = 10$), none distinguishes D2 and D3, and one (*Tbx3*) differentiates D3 from D4 in all three species (Fig. 4a).

Overall, data from these four species do not support the hypothesis that amniote digits have conserved developmental identities. The exception appears to be D1, which likely had a distinct developmental program in the most recent common ancestor of amniotes. We further tested whether D1 has a conserved gene expression profile by sequencing RNA from developing human fore- and hindlimb, which were partitioned into D1 and the posterior digital plate (D2–5). Of the ten genes identified above as differentiating D1 and D2, six show conserved

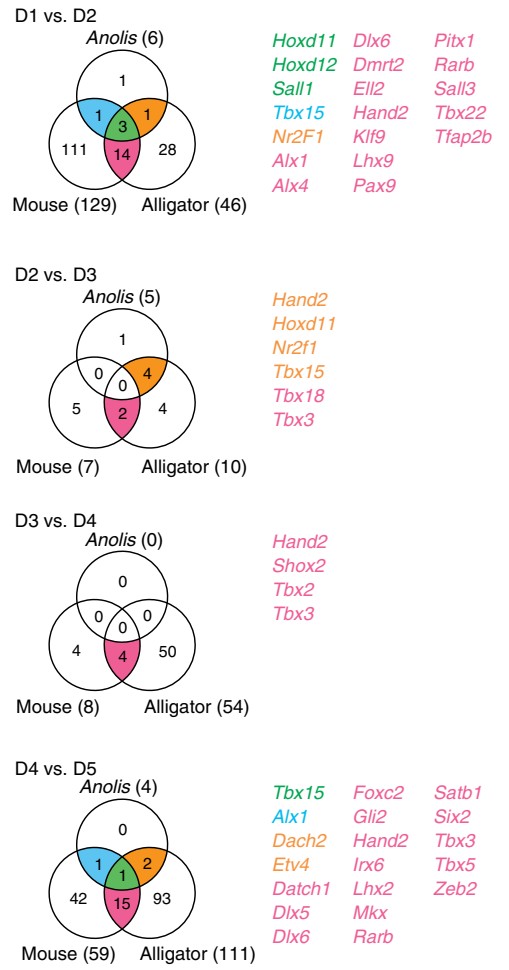

**Fig. 3** Few genes are differentially expressed at the same position between pentadactyl limbs. Venn diagrams of one-to-one orthologous transcription factors genes for mouse, alligator, and *Anolis* that were identified as differentially expressed between adjacent digits with an FDR threshold of 0.05

particular FDR threshold being reached in all species. Specifically, we consider the genes identified as differentially expressed in one species between adjacent digits (e.g., in mouse, 129 transcription factor genes are identified between D1 and D2). Then we ask how expression fold change between the two digits in the original species compares with expression fold change of the same genes and also a set of randomly selected genes of similar expression levels in other species. To make these comparisons, we calculated Pearson's correlation of the fold changes between the original genes vs. each of the two gene sets (orthologs and random genes) in other species. Results of this approach broadly mirror those described above.

Among the pentadactyl limbs sampled, genes differentially expressed between D1 and D2 behave consistently between species and can be distinguished from random genes, and comparisons of the more posterior digits do not clearly distinguish orthologs from random genes, (Supplementary Fig. 5a–d). If chicken hindlimb is considered instead of the *Anolis* forelimb, we again obtain strong support for conserved behavior of genes at the position D1 and D2, weaker support for conserved gene behavior between D2 and D3, and comparisons at the position D3 and D4 do not clearly distinguish orthologs from random genes (Supplementary Figure 5e-g). Thus, testing for genes that are differentially expressed at the same position can recover genes that behave consistently across species (i.e., *Tbx15* between D4 and D5 among pentadactyl limbs, and *Tbx3* between D3 and D4 between mouse, alligator, and *Anolis*), while comparisons of all genes differentially expressed for these species might not show evidence of broadly conserved profiles. Conversely, while we might obtain modest evidence for shared behavior among differentially expressed genes (i.e., between digits D2 and D3 among mouse, alligator, and chicken), there might be no individual genes recovered as differentially expressed among the taxa at that position. However, both types of comparisons between digits D1 and D2 paint the consistent picture that D1 exhibits a shared digit identity across these limbs.

**A core set of digit patterning genes**. Given our result that the gene expression profile of digits is evolutionarily dynamic, we next tested whether a conserved set of genes might pattern amniote autopods, albeit in different spatial patterns. Specifically, we reanalyzed transcriptomic data of mouse, alligator and *Anolis* forelimbs and chicken hindlimb, conducting ANOVA to test for genes that were differentially expressed between any two digits in the limb, not just adjacent digits. This analysis recovers genes that are differentially expressed between some digits in the limb, but it does not indicate between which digits a gene is differentially expressed. The number of differentially expressed transcription factor genes differs greatly among species: 356 in mouse, 377 in alligator, 34 in *Anolis*, and 144 in the chicken hindlimb (FDR < 0.05, Fig. 5a). This is consistent with previous results (above) that showed the *Anolis* forelimb to be more homogeneous than other sampled limbs. Therefore, we focused on transcription factor genes that are one-to-one orthologous between mouse, alligator, and chicken and identified a set of 49 genes that are differentially expressed in these three limbs (Fig. 5b). We call these "conserved differentially expressed genes" (CDEGs). The expected number of overlapping genes among these sets by chance alone is 7.57, and the probability of observing an overlap of 49 genes or more by chance is $<10^{-6}$ (binomial test). Thirteen of the CDEGs are included in the list of limb patterning genes sensitive to *Shh* signaling[26]. To assess whether this gene set is biologically meaningful, we performed HCA and PCA on the samples of each species using the 49 CDEGs. In *Anolis*, we considered the subset ($n = 42$) that are one-to-one orthologs across all four species. In

patterns of expression change at this position: in all limbs sampled *Hand2, Hoxd11, Hoxd12,* and *Tfap2b* are more highly expressed in D2 than D1, and *Alx1* and *Pax9* are more lowly expressed in D2 than D1 (Fig. 4b).

T-box family genes are predicted to regulate the identities of posterior digits[15]. Our data provide some support for the hypothesis that this function is conserved across amniotes (Fig. 4c). *Tbx2*, which was previously shown to regulate posterior digit identity in the chicken hindlimb[15], shows divergent patterns of expression in the posterior digits of other species. *Tbx3* differentiates D3 from D4 in mouse, alligator, and chicken hindlimb, and the likelihood that it was recovered by chance alone is $7.2 \times 10^{-6}$ (binomial test); however, it is not differentially expressed at this position in *Anolis* forelimb. *Tbx15* differentiates D4 from D5 among pentadactyl limbs (Fig. 3), and the likelihood that it was recovered by chance alone is $1.9 \times 10^{-5}$ (binomial test).

Analyses aiming to identify genes that are conserved and differentially expressed at a particular position within the limb (e.g., between D1 and D2 in mouse, alligator, and *Anolis*) can be affected by the threshold stringency of the false discovery rate (FDR). Binomial tests, as presented above, are one means of accounting for this. We present a second strategy for assessing whether genes identified as differentially expressed in one species behave similarly in other species that does not depend on a

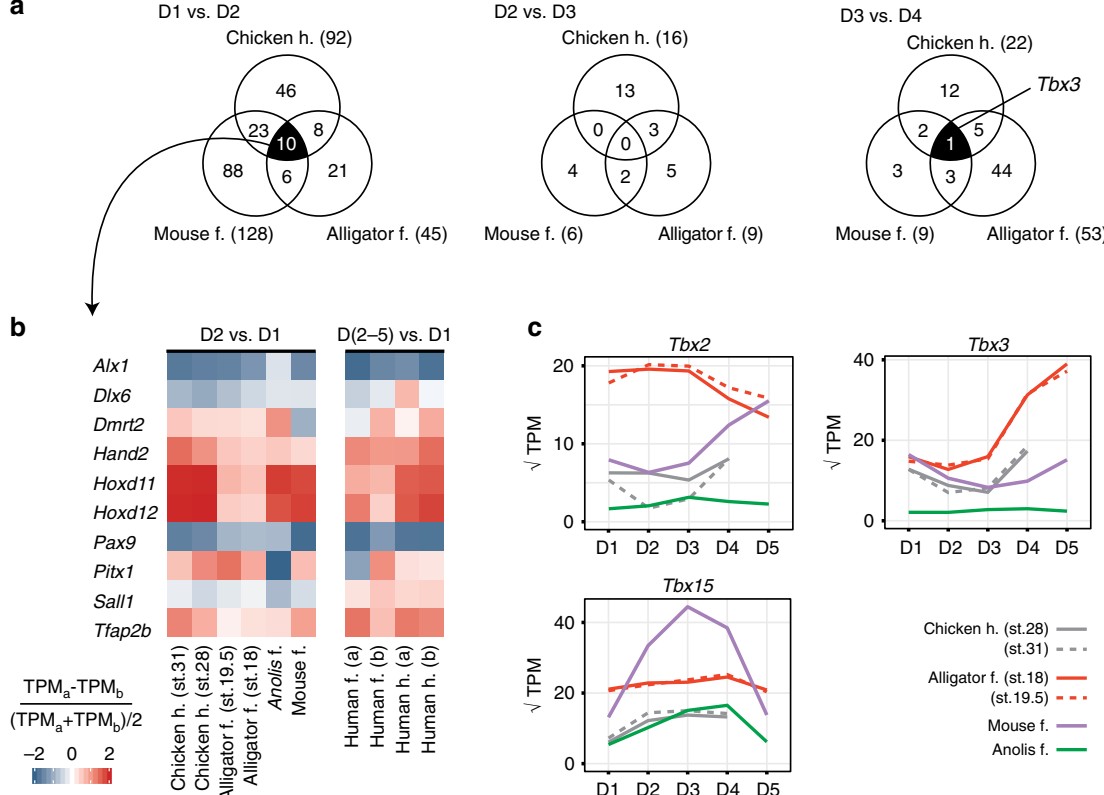

**Fig. 4** D1 has a unique, conserved gene expression profile across amniotes. **a** Venn diagrams of one-to-one orthologous transcription factors genes of mouse, alligator, and chicken that are differentially expressed between adjacent digits (FDR threshold of 0.05). **b** Heatmap showing relative expression of genes in D2 and D1. Human transcriptomic data provides additional support for the hypothesis that D1 has a conserved developmental identity across amniotes. **c** Expression levels of *T-box* family genes across the autopod. Transcript per million (TPM) values presented in panels **b** and **c** calculated from the gene list of one-to-one orthologous genes between mouse, alligator, *Anolis*, chicken, and human. Source data for panels **b** and **c** are provided as a Source Data file

combination, CDEGs can produce unique expression profiles of each digit within a limb (Fig. 5c) and show patterns similar to those generated by analyses of known limb patterning genes (Fig. 1b–d, Supplementary Fig. 3c).

Analysis of amniote limbs showed that targeted gene lists generated either experimentally (i.e., known limb patterning genes[26]), by gene ontology (i.e., all transcription factors[27]), or statistically (i.e., 49 CDEGs), can reveal distinct gene expression profiles among digits of a limb, which are not observed in the full transcriptome. The spatial digit expression profiles of these genes, however, is species specific. In light of these results, we reevaluated the frame shift hypothesis of bird wing origin[16].

**Reevaluating the frame shift hypothesis**. The frame shift hypothesis predicts that the three digits of the adult avian forelimb, which we refer to here as D2, D3, and D4 according to their developmental position[17–22], will express the developmental programs observed in the digits D1, D2, and D3 of other limbs[16]. This hypothesis was tested previously by analyzing the transcriptomes of chicken fore- and hindlimb digits[7]. That study found correspondence between forelimb D2 and hindlimb D1, consistent with the frame shift hypothesis. However, correspondence of more posterior digits was not detected[7].

We reanalyzed published transcriptomic data of digits from the chicken forelimb[7] and compared them to digits of the chicken hindlimb. Surprisingly, when the 49 CDEGs are considered, gene expression profiles of forelimb digits D2, D3, and D4 correspond to hindlimb digits D1, D3, and D4, respectively (Fig. 6a). Analyses of transcription factor genes and known limb patterning genes

show a consistent pattern (Supplementary Fig. 6). Similarity between the posterior two digits of the chicken fore- and hindlimb (D3 and D4 in each limb) can also be observed in the expression patterns of numerous individual genes that are known to be involved in the patterning of digits (Fig. 6b).

To assess whether spatial gene expression profiles can be conserved between the fore- and hindlimbs of a species, even when they differ in digit number, we performed in situ hybridization in alligator. We evaluated expression of *Tbx2*, *Tbx3*, and *Sall1*, three transcription factor genes identified as differentially expressed between alligator forelimb D3 and D4. In situ hybridization confirms their expression in the posterior interdigital mesenchyme (Fig. 6c) and shows conserved positional expression patterns for *Tbx3* and *Sall1* between the forelimb and hindlimb. It is unclear whether the pattern also holds for *Tbx2*, where difference in expression level detected from RNA sequencing appear to reflect the proximodistal extent of gene expression.

We also tested the frame shift hypothesis by comparing the limbs of chicken to the pentadactyl forelimbs of other species. For each pentadactyl species, PCA were run using the CDEGs (49 in mouse and alligator, and 42 in *Anolis*), and chicken samples were projected into the reference PCA plane as supplementary observations. CDEGs were used because they can produce digit-specific expression profiles for mouse and alligator forelimb and chicken hindlimb, and because these patterns are reflective of more inclusive gene lists, as described above. These projections show that the digits D2, D3, and D4 of the bird wing consistently fall into regions of the PCA plane corresponding to the digits D1,

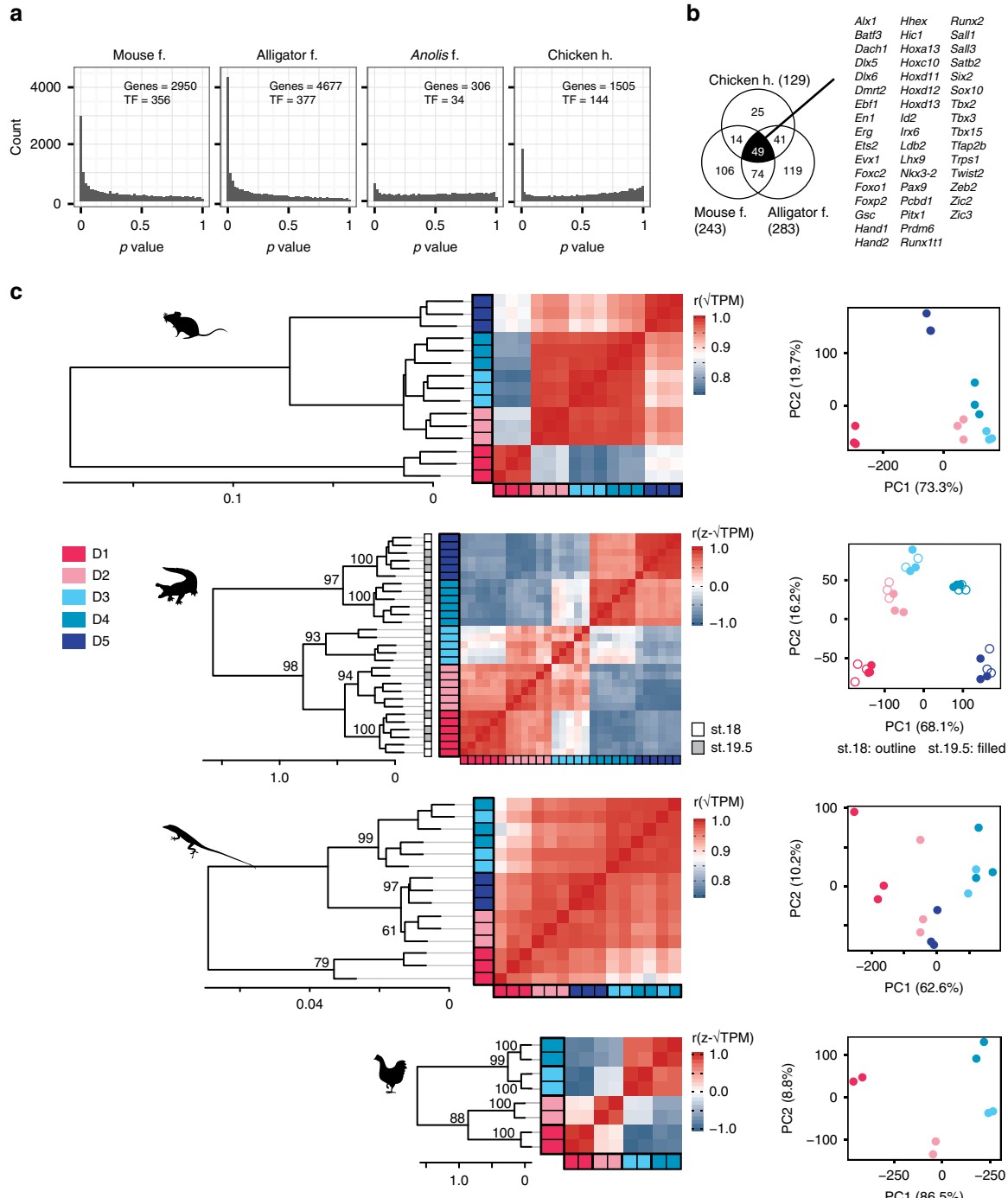

**Fig. 5** A conserved core set of digit patterning genes in amniotes. **a** ANOVA comparing all combination of digits within limbs. The number of genes identified as differentially expressed at a FDR threshold of 0.05 are noted in each panel as "genes." The number of transcription factors among these are noted as "TF." **b** Venn diagram showing one-to-one orthologous transcription factor genes that are differentially expressed across mouse, alligator, and chicken limbs. **c** HCA and PCA show these genes can recover patterns of digit correspondence similar to analyses of limb patterning genes and transcription factors. Alligator and chicken illustrations reproduced with permission by Michael Richardson. *Anolis* illustration by Sarah Werning without modification (license [https://creativecommons.org/licenses/by/3.0/])

D3, and D4 of other limbs. Although it is difficult to differentiate D3 and D4 expression in all species, it is clear that D3 of the chicken forelimb does not correspond in its expression profile of these genes to the D2 of the other limbs sampled (Fig. 6d).

Skeletal tissues in the three digits of the adult avian wing (D2–D4) do not derive from *Shh* expressing cells[24,25]. This pattern is consistent with digits of positions D1–D3 in other limbs and can be regarded as evidence in favor of the frame shift hypothesis. Our data are incongruent with such diagnoses of digit developmental identity for the second and third digit in the avian wing, but confirm previous identification of the first avian wing digit as homologous to D1 in the pentadactyl ground plan[7,23]. Thus, according to the prediction that core regulatory networks of transcription factor genes responsible for developmental identity are expressed in the posterior interdigital mesenchyme, we suggest that *Shh* expression is not a conserved

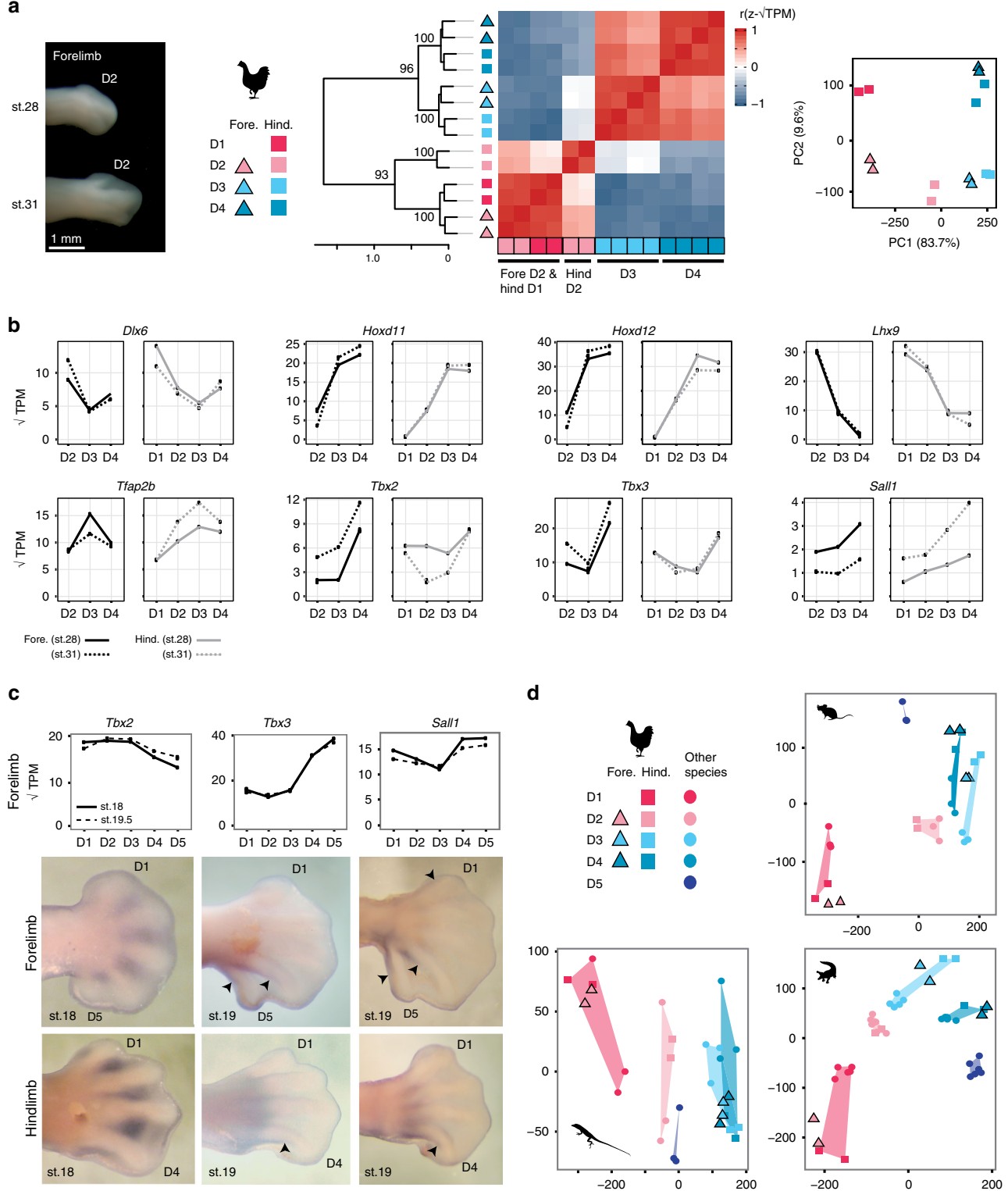

**Fig. 6** Digits D2, D3, and D4 of the chicken forelimb correspond to D1, D3, and D4 of other limbs. This pattern is observed by **a** HCA and PCA of CDEGs for digits of chicken wing and hindlimb. **b** Expression levels of individual transcription factor genes in chicken. **c** In situ hybridization of alligator embryos confirms that the position of differential expression of transcription factors can be conserved between fore- and hindlimbs that differ in digit number. Arrowheads indicate distal-most expression along a developing digit. **d** Projection of chicken digit data upon PCA of CDEGs for mouse, alligator, and *Anolis*. TPM values shown in panels **b** and **c** calculated from the gene list of one-to-one orthologous genes between mouse, alligator, *Anolis*, chicken, and human. Source data for panel **c** are provided as a Source Data file. Alligator and chicken illustrations reproduced with permission by Michael Richardson. *Anolis* illustration by Sarah Werning without modification (license [https://creativecommons.org/licenses/by/3.0/])

marker of developmental digit-identity in limbs with highly reduced digit number.

## Discussion

Serial homologs are repeated body parts, generated by a common developmental program. In the case of digits, chondrogenic condensations are generated by a reaction-diffusion Turing-type mechanism[32,33]. Serial homologs can be developmentally identical (homomorph parts) or they can assume distinct developmental identities through the differential expression of regulatory genes (paramorph parts)[34]. The degree to which serial homologs are individuated can be difficult to assess from morphology alone, because the same developmental program can lead to different morphological outcomes depending on the developmental environment[35,36]. However, detailed analyses of gene expression and regulation can identify developmentally individualized body parts.

In this study, we performed a comparative analysis of whole genome expression data to test the hypothesis that digits have conserved developmental identities. In interpreting our data, we acknowledge that gene expression does not demonstrate gene function. Nevertheless, a lack of differential gene expression between digits is evidence of a lack of developmental individuation, and a high level of differential expression (particularly in transcription factor and signaling genes) is evidence for distinct gene regulatory states.

We studied gene expression in the developing digit and the interdigital mesenchyme after digit condensation. This approach was guided by previous experimental work. It was demonstrated in the chicken hindlimb that genes expressed in the interdigital mesenchyme regulate digit-specific morphologies[13] and that this signaling, in the phalanx-forming region, is active between stages 27 and 30[14]. Here, we analyzed the expression profiles chicken hindlimb digits of stages 28 and 31 and showed that expression profiles of limb patterning genes and transcription factor genes are stable over this developmental window (Figs. 4b, c, 6b and Supplementary Fig. 3b, c). Thus, signals pertinent to digit patterning continue to be expressed at late stages of limb development, even after phalanges have formed. Analyses of alligator show a consistent pattern: between stages 18 and 19.5, the expression patterns of limb patterning genes and transcription factor genes are stable as assessed by HCA, PCA, and in profiles of genes of interest (Fig. 1c, Supplementary Fig. 2 and Figs. 4b, c, 6c). Although not all species were sampled at multiple time points, we argue on the basis of these comparisons in chicken and alligator that it is unlikely our conclusions on the evolution and development of digit identity are biased by temporal dynamism in gene expression within the developmental window studied here.

Our analyses show that patterns of regulatory gene expression in digits are evolutionarily dynamic (Fig. 7a). The developmental identities of digits are evolving across amniotes and can be lineage-specific. The exception is a conserved developmental identity that characterizes the D1 of mouse, alligator and Anolis forelimbs, chicken hindlimb, and human fore- and hindlimbs (Fig. 4b). This digit identity is unlikely to be an edge effect (i.e., merely a corollary to which digit occupies the most-anterior position in a limb). In the rabbit hindlimb, which has lost the digit D1, this developmental identity is not observed in D2, despite that digit now occupying the anterior-most position in the limb[37]. In addition, in the hindlimb of Silkie chicken mutants, which have additional anterior digit on their foot, developmental identity is preserved in the digit of the morphology of the native D1, despite that digit no longer occupying the anterior-most position in the limb[23].

In contrast to D1, we do not find support for conserved digit identities in the more posterior digits. Among the pentadactyl limbs we studied, no genes consistently distinguish the median digits (D2, D3, and D4) from one another. And when we consider the chicken hindlimb rather than Anolis, because similarity among Anolis digits might be secondarily derived, we find no gene differentiates D2 and D3, and only one gene (Tbx3) differentiates D3 and D4. There is limited evidence for a conserved developmental identity for digit D5. A single gene (Tbx15) is differentially expressed between D4 and D5 among mouse, alligator and Anolis, however more genes are shared between just mouse and alligator (Fig. 3, Supplementary Fig. 4b).

Our analyses also identified a core set of regulatory genes, which we call CDEGs, that are differentially expressed among digits, although the exact pattern of gene expression among digits differs between species (Fig. 5). We propose that the CDEGs represent a "digit differentiation tool kit" deployed for the individuation of different sets of digits in different lineages, depending on the evolutionary history and the adaptive needs of the species. Between mouse and human, 28 of the 49 CDEGs have demonstrated roles in patterning distal limb skeleton (Supplementary Table 1). Of the CDEGs, only 15 are differentially expressed across the Anolis forelimb. This homogeneity of expression in Anolis appears to be a derived condition among the taxa sampled, as it is unlikely that the other 34 CDEGs reflect homoplasy between mammals and archosaurs.

In Anolis most fingers, though they differ in number of phalanges, lack developmental individuality and, thus, appear to be homomorphic. We considered a number of alternative, non-biological explanations for the unique Anolis pattern; however, these do not adequately explain homogeneity in the data. For example, it is possible that the limbs were sampled at too-late a stage, after signals pertinent to digit patterning ceased to be expressed. We regard this explanation as unlikely because, as discussed above, in limbs sampled at multiple time points gene expression profiles are stable over broad developmental window, through late stages of development. Another possible alternative explanation is that variance among Anolis samples is greater as compared with other data sets, and that this diminished our ability to detect differentially expressed genes. We assessed this possibility in two ways. First, we repeated all differential expression analyses considering only the two most highly correlated samples of each digit for mouse, alligator and Anolis, which consistently had correlation values above 0.99 (Supplementary Fig. 7a). Results of these two-sample comparisons are consistent with analyses of all three samples (e.g., compare Fig. 3 and Supplementary Fig. 7b), indicating that the unique Anolis pattern is not an artifact of sample quality.

Second, we evaluated the dispersion values of our samples. Dispersion is a measure of variance among samples that is calculated by the software edgeR. This parameter affects the sensitivity of differential expression analyses (e.g., a set of samples with high dispersion will have low sensitivity in tests of differential expression), and it can be impacted by specimen pedigree[38]. Anolis embryos were collected from nonsiblings, whereas mouse and alligator samples were collected from siblings. As expected, the mean dispersion value of Anolis samples is greater than either mouse or alligator (Supplementary Fig. 8). The Anolis mean dispersion value is consistent with other data sets in which samples were collected across a population[38]. However, such differences in dispersion cannot explain the unique Anolis pattern. Chicken hindlimb digits, which were also collected from nonsiblings and have dispersion values comparable to Anolis (Supplementary Figure 8), show patterns of differential expression comparable to mouse and alligator (Figs. 2, 5a and Supplementary Fig. 4a). Thus, neither timing, sample quality, nor

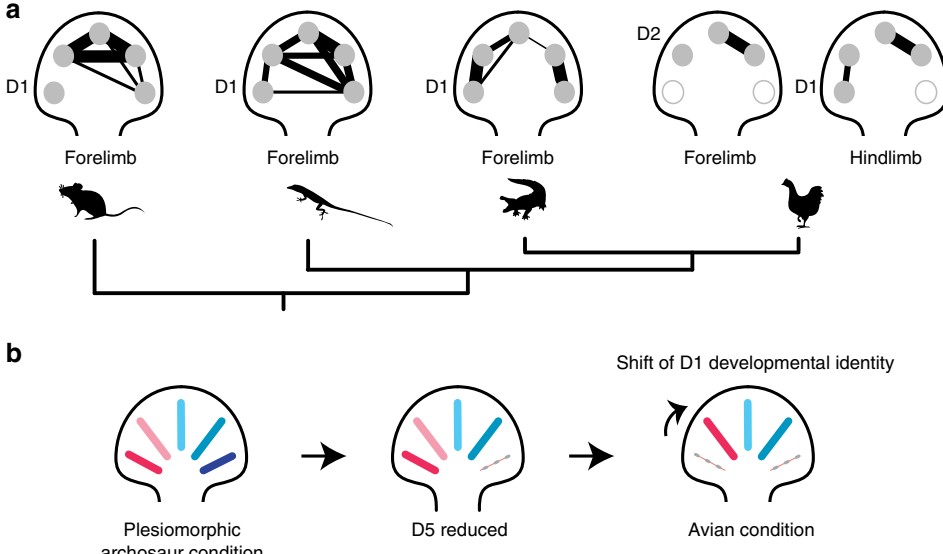

**Fig. 7** The evolution of digit gene expression in amniotes is highly dynamic. **a** A phylogeny of the taxa sampled by this study and schematic graphs summarizing the relative similarity of digits within limbs, where connections and line thickness reflect degree of similarity in gene expression profiles. **b** Schematic of a limited frame shift model for evolutionary origin of the avian wing in which the developmental identity of D1 was translocated to position D2. Alligator and chicken illustrations reproduced with permission by Michael Richardson. *Anolis* illustration by Sarah Werning without modification (license [https://creativecommons.org/licenses/by/3.0/])

pedigree appears sufficient to explain the *Anolis* data. It appears that homogeneity of gene expression among the digits reflects biological reality, and that digits in this lineage have undergone secondary homogenization. Other lineages might have similarly experienced loss of digit identities (e.g., ichthyosaur forelimbs), and the secondary homogenization of paramorphic serial homologs has been described in other anatomical systems (e.g., the homodont dentition in cetaceans[39] and the snake vertebral column[40]).

Finally, we reassessed the homology of fingers in the bird wing and obtain the result that the three digits reflect a combination of translocated and conserved digit identities. The anterior-most digit in the chicken wing, although it develops in position D2[17–22], exhibits a gene expression profile seen in the D1 of other limbs; this is consistent with previous studies and the frame shift hypothesis[7,23]. The gene expression profiles of the posterior wing digits (D3 and D4), however, do not show evidence of translocation. This is observed most clearly by comparison to the hindlimb of the chicken, with the pattern recovered when three different gene lists are considered (transcription factors, limb patterning genes, and CDEGs). As discussed above, although we cannot diagnose conserved gene expression profiles for the digits D3 and D4 across amniotes, we obtain indirect evidence for a correspondence of avian digits to the digits D1, D3, and D4 of other amniote limbs (Fig. 6d). The possibility of a 1, 3, 4 pattern of digit identity in the bird wing has been proposed previously[41] on the basis of experimental studies[42]. Still, this pattern of correspondence is surprising. It challenges the predominant hypotheses of digit identity and suggests an alternative scenario for how limb development evolved in the lineage leading to Aves (Fig. 7b). Significantly, it indicates that diagnoses of digit identity from the paleontological record and hypotheses of digit identity based upon gene expression profiles have a more complex relationship than previously anticipated.

The frame shift hypothesis is an integrative model. It aimed to explain an apparent incongruity between paleontological and neontological data sets by providing a developmental account for evolutionary transformation rooted in a mechanistic model of

homology. Our results show that any such integrative model will be more complicated than previously presumed. Moving forward, we recommend systematic reappraisal of phalangeal and metacarpal characters along the avian stem. It has been proposed that patterns of digit reduction in theropods might be more complex than is generally assumed[41]. For example, study of the ceratosaur *Limusaurus* led to the hypothesis that in basal tetanurans metacarpal characters correspond to identities 2-3-4, while phalanges have identities 1-2-3[43], although specifically how this taxon informs the plesiomorphic avian condition has been contested[44]. In addition, we recommend continued, broad taxonomic sampling in studies of limb development. Building expanded, comparative data sets will allow for documentation of homoplasy between species and between the fore- and hindlimbs, which could impact hypotheses of digit identity presented here. Sampling should also be extended to earlier timepoints to test for the possibility that developmental identities might be detectable at earlier stages. Finally, additional functional genetic studies are required to understand how digit-specific phenotypes are regulated and to test the hypothesis that CDEGs play privileged roles in establishing gene regulatory states in the interdigital mesenchyme.

The question of how to diagnose the digits of the avian wing is among the oldest in comparative morphology[3,45]. This study tests several assumptions that underlay many contemporary studies of the homology and developmental identity of digits. Indeed, it is the first to comparatively analyze the full gene expression profiles of digits of different species. Such data, and a willingness to consider hypotheses that previously might have been regarded as heterodox, are required for the testing and refinement of integrative theories on the nature of limbs.

## Methods

**Tissue and taxon sampling strategy**. Limbs of each species were sampled after digital condensations have formed and after interdigital webbing has begun to reduce. RNA was extracted from digits and their associated posterior interdigital webbing following the dissection strategy shown in Fig. 1a of Wang et al.[7]. A summary of the taxa sampled and the tissues collected is shown in Supplementary

Fig. 9. Investigators were not blinded to the group allocation during the experiment or when assessing outcomes.

Protocols for mouse care and euthanasia were approved by Yale University's Institutional Animal Care and Use Committee (protocol #2015-11-483). Protocols for *Anolis* care and breeding were approved by Loyola University's Institutional Animal Care and Use Committee (protocol #1992). All methods were performed in accordance with relevant local guidelines and regulations.

**Alligator mississipiensis**. Fertilized eggs were collected from six nests of wild individuals at the Rockefeller Wildlife Refuge in Grand Chenier, LA (USA) in July 2015 by Dr Ruth Elsey and colleagues. Eggs were marked with pencil to indicate the side that was facing upwards in the nest so that embryos would not be injured by rotation of during transfer. Eggs were transported to Yale University in mesh wire boxes containing original nesting material, and they were incubated in a temperature-controlled room at 32 °C. Eggs were placed on a plastic rack, surrounded with original nesting material. Racks were suspended four inches above the bottom of a 10 gallon aquarium. The base of the aquarium was filled with three inches of water, which was heated to 90 °F with a submerged aquarium heater. The top of the tank was covered with plexiglass perforated with 1 cm diameter holes to allow for airflow. Humidity within the tank was maintained at 90%.

Embryos were collected at Ferguson[46] st.18 and 19.5. The left and right limbs of ten individuals were dissected at each stage. For each stage, individual samples were of a single nest and, therefore, at least half-siblings[47]. Embryos were extracted under sterile, RNAse-free conditions. Individual digits and the associated posterior interdigital webbing were dissected with fine scissors and forceps and placed immediately in room temperature RNAlater (Sigma-Aldrich). Digits were pooled into a single vial ($n = 20$ digits) and divided into four samples of five randomly selected digits. RNA was extracted from each sample with TRIzol (Thermo Fisher Scientific)[48]. For digit D3 stage 18, two of the extractions yielded too little RNA for sequencing approach described below; therefore, there are only two replicates of this sample type. RNA quality was assessed using an Agilent Technologies 2100 Bioanalyzer, and samples with RIN scores above 8.5 were submitted for sequencing at the Yale Genome Sequencing Center. Sample size (three replicates per sample type) was selected for downstream differential expression analyses, according to references[29,30]. To generate strand-specific polyadenylated RNA libraries, samples were processed as follows: Approximately 500 ng of RNA was purified with oligo-dT beads and the mRNA recovered was sheared by incubation at 94 C. First strand synthesis was performed with random primers, and then second strand synthesis was performed with dUTP to generate strand-specific libraries for sequencing. cDNA libraries were end paired, A-tailed adapters were ligated, and the second strand was digested with Uricil-DNA-Glycosylase. qRT-PCR was performed using a commercially available kit (KAPA Biosystems) to confirm library quality, and insert size distribution was determined with Agilent Bioanalyzer. Samples were multiplexed on an Illumina Hiseq 2000. Each sample was sequenced to a depth of ~50 million reads (single-stranded, 75 base pair length).

Reads were mapped to the American alligator genome assembly (allMis0.2) with genome assembly described by Green et al.[49]. Sequenced reads were mapped to the genome using Tophat2 v2.0.6 on Yale University's Ruddle computing cluster. In Tophat2, reads were first mapped to the transcriptome, and the remaining reads were then mapped to the genome. Mapped reads were assigned to genes with *HTSeq v0.5.3p*[50], which was implemented with Python v2.7.2. In HTSeq, we required that reads be mapped to a specific strand, and to account for reads that mapped to more than one feature, we ran with the setting "intersection-nonempty."

**Mus musculus**. Mice embryos (E13.5) were collected from a pregnant female of the strain C57BL/6J (Jackson Laboratories) in accordance with Yale IACUC #2015-11-483. The female was pregnant with nine embryos. Digits from the left and right forelimbs of each individual were dissected as described for alligator and pooled. From these 18 digits, RNA was extracted for three batches of five digits each. RNA extraction and sequencing methods are the same as described above for alligator, with the exception of sequencing depth (30 million reads were obtained for each mouse sample). Sequenced reads were mapped to the mouse genome assembly GRCm38 with Ensembl annotation v85 and the same Bowtie2 and HTSeq settings as described for alligator.

**Anolis carolinensis**. Animals were bred according to published protocols[51] and in accordance with Loyola University's IACUC protocol #1992. Fertilized eggs were collected and transferred to petri dishes containing vermiculate moistened by equal mass water. Embryos were shipped to Yale University and incubated in a Digital Sportsman Incubator (No. 1502) at 26 °C. Tissues were extracted and dissected according to methods described for alligator. Stage 10[52] embryos were sampled, and RNA was extracted using the Qiagen RNeasy Micro Kit. RNA quality was assessed using with a BioAnalyzer. Samples with RIN scores above 9.0 were submitted for sequencing at the Yale Genome Sequencing Center. The RNAseq library was prepared with the Clontech's Ultra Low V4 kit (cat# 634890). Each sample was sequenced to a depth of ~30 million reads (single-stranded, 75 base pair length). Sequenced reads were mapped to the *Anolis* genome assembly (AnoCar2.0, GCA_000090745.1) with Ensembl annotation v85.

**Gallus gallus**. Published transcriptomes of the digits of the fore- and hindlimbs of chicken[7], were mapped to the newest chicken genome version (GalGal5.0) with Ensembl annotation v86 following analytic methods described above for alligator.

**Homo sapiens**. Three individuals of Carnegie stage 18[53] were donated to Yale University's Medical School. The fore- and hindlimbs were sampled, and the anterior-most digit and its posterior interdigital webbing were dissected from the posterior digital plate. Dissections were performed and RNA was extracted with an RNEasy Kit (Qiagen) and prepared for sequencing with the Illumina mRNA-seq Sample Prep Kit. Samples were sequenced on an Illumina GA IIx (single-stranded, 35 base pair length)[54]. Limbs at this stage are similar to E12.5 of mouse[55]. Sequenced reads were mapped to the human genome assembly GRCh37 with Ensembl annotation v82 using the same Bowtie2 and HTSeq settings as described for alligator.

**Hierarchical clustering analysis**. To estimate relative mRNA abundance, we calculated transcripts per million (TPM)[56] for the genes of a given gene list (i.e., full transcriptome, transcription factors, limb patterning genes, CDEGs). The TPM measurement standardizes for sequencing depth and transcript length. If multiple transcripts are described for a gene, then the median transcript length was used to calculate TPM; these lengths are available in the file Supplementary Data 2. TPM measures were normalized by a square root transformation, and hierarchical clustering was performed on the normalized TPM data with the R package "pvclust"[57]. Clusters were generated from the correlation-based dissimilarity matrix using the average-linked method. Adjusted uncertainty values were calculated from 1000 bootstrapping analysis.

If analyses involved comparisons between developmental stages or limbs, a bulk correction was performed with a mean transformation (i.e., mean centering)[7]. In these instances, Pearson's correlation coefficients range from [−1:1], rather than from [0:1]. Negative correlation values arise because after bulk correction, a gene's expression values is negative for samples with a sqrt(TPM) value less than the mean sqrt(TPM) value of that gene among all samples of the bulk. Bulks were comprised of all samples from a particular stage or all samples of a particular limb.

**Principal component analyses**. PCA were performed using the "prcomp" function in R for various gene lists using square root TPMs as normalized measures of relative mRNA abundance. As with HCA, if analyses included samples from multiple stages or from different types of limbs, a bulk correction was performed with a mean transformation (i.e., mean-centering). Bulks were comprised of all samples from a particular stage or all samples of a particular limb. Loading values for samples in PCAs and also bootstrap values, which were calculated using the with the bootPCA' function of the bootSVD package[58] with centerSamples = True and 1000 bootstrap samples), are provided in the file Supplementary Data 3.

**Differential expression testing of adjacent digits**. EdgeR (Release 3.1)[29,30] was used to test for differential expression of adjacent digits (e.g., D1 vs. D2) of mouse, alligator, *Anolis* and chicken. We used function glmFit and glmLRT in EdgeR, which implemented a generalized regression model for differential expression test. Specifically, in alligator and chicken PCA and PCA revealed stable grouping of samples by digit number across stages. Therefore, subsequent analyses consider these data from two stages simultaneously.

In *Anolis*, pairwise testing of samples revealed nonconventional $p$ value distribution, with a decrease near zero and sometimes a bump near 0.5. Because correlation between replicates was lower than what was observed in either mouse or alligator (Supplementary Fig. 7), and because PCA of the full transcriptomes revealed two major clusters of data that did not correspond to biological phenomena (Supplementary Fig. 1c), we corrected for the artifact of the nonbiological clusters by including the first principle component in the regression model in EdgeR. Analyses were also run without this PC1 correction. Results presented in the manuscript are robust to both analytic approaches, although PC1 correction results in discovery of slightly more differentially expressed genes for a given FDR. (e.g., compare Fig. 3 and Supplementary Fig. 10).

Following analyses of differential expression, multiple hypothesis testing was accounted for by adjusting $p$ values following the Benjamini–Hochberg method[28]. We also considered a second correction method, the $q$-value of Storey[59]. The major results presented in the study are robust to both methods. Although the Storey method uniformly called more genes as significant at the FDR threshold of 0.05, the same genes are recovered in the center of Venn diagrams (Figs. 3, 4a, and 5b).

**Correlation of fold change in orthologs and random genes**. To assess whether genes differentially expressed at a given position in one species are behaving similarly in other species, we compared the relative fold change of these genes to random genes of similar expression level in the other species. For example, between D1 and D2 of alligator 46 genes are recovered as differentially expressed among the one-to-one orthologs of the three pentadactyl species sampled (FDR threshold of 0.05) (Fig. 3). For these genes, we calculated the fold change in TPM according to the equation in Fig. 4b for all three species (e.g., mouse, alligator, and *Anolis*). Next, for each of the 46 genes, we identified the gene most similar in its TPM value at the position of the anterior digit (e.g., for the comparisons of D1 vs D2, we matched

the TPM of D1) among the one-to-one orthologous transcription factor genes for the two other species. Then, we calculated to Pearson's correlation for the vector comprised the gene fold changes from the original species (alligator) and the orthologs of the other species (mouse and *Anolis*). We also calculated Pearson's correlation between gene fold changes from the original species (alligator) and the random list of random genes of similar expression level for each of the other species. This was repeated for each the three species, and if a limb was sampled at multiple time points, for each time point. Finally, to assess whether at a given position orthologous genes could be distinguished in their behavior to random genes, we compared the means of these correlations using two tests: *t*-test and a Mann–Whitney *U* test.

**Differential expression testing between all digits**. EdgeR was also used to test for genes that differed between any combination of digits within a limb. This was done by specifying multiple coefficients to function glmLRT. As with the pairwise tests, we considered both stages of alligator and chicken simultaneously and included the first principle component in the regression model for *Anolis*. Genes identified as CDEGs for each species are available as Supplemental Information.

**Transcription factors**. To identify transcription factor genes, we utilized a published atlas of human and mouse transcription factors[27]. The published Entrez gene IDs were matched with human Ensembl gene IDs in Ensembl assembly v85 using BioMart (*N* = 2183). To recover the species-specific lists of transcription factors, two approaches were taken. In mouse and *Anolis*, orthologous genes were identified using BioMart's orthology predictions for Ensemble assembly v85. In alligator and chicken, because these species were analyzed using different assembly builds, orthology was determined by matching gene symbols to those of human from Ensembl assembly v85. By this approach, we recovered transcription factors for each species as follows: 1838 in mouse, 1563 in *Anolis*, 1455 in Alligator, and 1217 in chicken. To identify human orthologs of select genes identified by differential expression analyses, we first used gene symbols and then confirmed that the ensemble IDs were consistent across Ensemble assemblies. Genes identified as transcription factors for each species are available as Supplemental Information.

To identify one-to-one orthologous transcription factor genes in mouse, alligator, and *Anolis*, we used BioMart to generate a list of one-to-one orthologous genes between mouse and *Anolis* in Ensembl assembly v85 that correspond to the published transcription factor Entrez IDs[31]. This gene list was then matched to alligator and chicken by gene symbol to recover transcription factor genes that are one-to-one orthologs in multiple species.

**Limb patterning genes**. A Ph.D. dissertation by Carkett[26] identified genes that are sensitive to *Shh* signaling by experimental perturbations and in silico analyses. From these studies, a summary list of genes that pattern the autopod was produced (pg. 172). This gene list includes transcription factor and signaling genes. These gene symbols were matched with each species to identify the subset of genes present in the genome assemblies that we considered for mouse (*n* = 151), alligator (*n* = 142), *Anolis* (*n* = 140), and chicken (*n* = 136). Genes identified as limb patterning genes for each species are available as Supplemental Information.

**In situ hybridization**. RNA from a stage 18 alligator limb was extracted, as described above for sequencing, and cDNA was generated with the High Capacity cDNA Reverse Transcription kit (Applied Biosystems). Primers were designed with Primer3[60] to amplify fragments of *Tbx2* (forward, GACCTTGGGCCTTCTCCTA C; reverse, GGGAGTTGTTTGGGGTTTTT), *Tbx3* (forward, ACCAGGGGTGGAT GAACATA; reverse, GCCCTAAAGCAGAGACATGC), and *Sall1* (forward, CTCACAGCTCAACAACCCAC; reverse, AAACCACCAGCCTCTACCTC). PCR products were purified with the QIAquick Gel Extraction kit (Qiagen) and cloned with the Topo TA cloning kit (Invitrogen) into the pCR 4-TOPO vector. Vectors were transformed into DH5α-T1 competent cells. Sense and antisense probes were prepared by linearizing plasmid with the restriction endonucleases *Not1* or *Pme1* and then transcribing the linearized product with T7 or T3 polymerase, respectively. Probes are labeled with digoxigenin (Sigma-Aldrich) and hybridized with the alligator embryos at 68 °C. Methods for in situ hybridization followed GEISHA Project miRNA Detection Protocol Version 1.1 [http://geisha.arizona.edu/].

**Reporting summary**. Further information on research design is available in the Nature Research Reporting Summary linked to this article.

## Data availability
The RNA-sequencing data for mouse, alligator, and *Anolis* (including counts of mapped reads), is available on Gene Expression Omnibus (GEO) repository under accession number GSE108337. Sequencing data for human limb samples is available through the database of Genotypes and Phenotypes (dbGaP) under study accession number phs001226.v1.p1. Supplementary Data 1 contains unique gene IDs corresponding to all gene lists described. Supplementary Data 2 contains median lengths of gene transcripts. Supplementary Data 3 contains bootstrap values for all PCA plots. Supplementary Data 4 contains mapped reads for chicken. The source data underlying Figs. 4b,c and 6c are provided as a Source Data file.

## Code availability
All code used for analyses is available on GitHub [https://github.com/ThomasAStewart/digit_identity_project].

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

## Acknowledgements

We thank the Wagner lab members and SE Newman for discussions on data interpretation. The human embryonic material was provided by the Joint MRC (G0700089)/Wellcome Trust (GR082557) Human Developmental Biology Resource. Work in the Wagner lab was supported by the John Templeton Foundation (Integrating Generic and Genetic Explanations of Biological Phenomena; ID 46919), by National Institutes of Health grant GM094780 (to J.P.N.), and by National Science Foundation grant 1353691 to G.P.W.

## Author contributions

T.A.S. and G.P.W. contributed project design and paper writing. T.A.S., T.J.S., J.L.C. and J.P.N. contributed data for analysis. T.A.S., C.L. and G.P.W. contributed analyses of data.

## Additional information

**Competing interests:** The authors declare no competing interests.

