## [Peer Review File · Nature Communications]

Reviewers' Comments:

Reviewer #1:

Remarks to the Author:

This manuscript examines the notion of digit homology by testing whether the digits have expression profiles that are conserved across amniotes. To this end the authors generate the transcriptome of each digit separately in three amniote species (mouse, Anolis and alligator). When considering a curated set of Shh responsive genes or the set of TFs, the HCA and PCA analyses show clustering by digit except in Anolis where clustering is very weak suggesting homogeneity across the autopod. These results are further confirmed by the analysis of the distribution of p-values from different expression analysis. However, very few of the differentially expressed genes between adjacent digits are conserved across species, also when considering the chicken instead of Anolis except for the comparison between D1 and D2. These findings support the already known unique profile of D1 across amniotes (here also shown for human) and indicate that the expression profiles of the rest of the digits are species specific. This late conclusion is novel and has strong significance for the field.

Another important finding of this study is that digits at position 3 and 4 in the chick wing cluster with D3 and D4 in the other amniote limbs shown no evidence of transformation of their identities. This leads the authors to suggest the alternative view that the digits in the bird wing are 1-3-4, as previously considered by Xu and Mackem. I think that this proposal needs a deeper discussion including its paleontology implications regarding the evolution of birds.

Overall I think that this is an interesting study with two main conclusions: i) the evolutionary dynamism of the expression profiles of D2-D5 and ii) the interpretation that the digits in the bird wing are 1-3-4. However, I think that several issues should be addressed before it can be presented to the community.

A weakness of this study is that it is restricted to a single stage in mouse and in Anolis. It is well known that the patterns of expression of many patterning genes, particularly those responding to Shh signaling, are very dynamic and this dynamism may impact the discovery of conserved expression patterns unless the stages between species and between samples are very well matched. This is important regarding the conclusions of the "evolutionary dynamism" of digits. In this regard it would be important to know the change in expression profiles between the two alligator stages and also it would be desirable to analyze another stage in mouse. In any case the temporal aspect should be widely discussed.

Along the same line, Hoxa13 is one of the 49 CDEGs identified. It has a dynamic pattern of expression starting at the posterior-distal tip and expanding rapidly to cover the complete autopod. It is considered the best marker for the autopod and clearly an important TF in the formation of the digits but it is more difficult to understand a differential expression among the digits unless this is fine/temporal differences in level of expression. The same applies to other genes in the 49 CDEGs set.

Another point that raises some concern is the range of distribution of Pearson's correlation coefficients, between 0.9-1 or even 0.98-1, two or three decimals appear to be needed to appreciate differences.

Please indicate the significance of the bulk correction performed when the analyses involved comparisons between developmental stages (Fig. 1, extended data Fig. 1 etc...) as well as the meaning of the negative correlations.

Minor:

- The rationale to select this particular stage for the study, condensations formed and interdigital spaces regressing, should be provided.
- I suggest changing the color used for D5 as, in some graphics, it is difficult to distinguish from that used for D1.
- Because the curated set of Shh responsive genes, the list of 159 genes considered as "limb patterning genes", has not been published (PhD source), I wonder whether it could be provided as extended data in this manuscript.
- I am curious about how many of the 49 CDEGs are included in the set of the 159 limb patterning genes. Some of the genes included in the 49 CDEGs are

Reviewer #2:

Remarks to the Author:

This is a very interesting study of digit transcriptome identity in 4 tetrapodes (plus human). The work and the interpretation are very careful, and this will be of interest to students of limb evolution, but more generally also for understanding the relation between anatomical homology and gene expression, a central question of Evo-Devo. I have a few comments on the methods and writing. It is especially important to make the study more reproducible. The authors lean on data and annotations from others, and must make theirs available for further studies.

Major comments:

As it is presented, the work is not reproducible. The authors need to make the following available:

- TPMs and counts as used for all samples;
- gene list of "known patterning genes";
- gene list of transcription factors as determined per species;
- gene list for the 49 CDEGs.

For all three gene lists, the authors should provide gene identifiers in all species, so that reproducibility of the work does not depend on future changes in orthology calls.

The "in-house protocol (available upon request from Yale Genome Sequencing Center) used to generate strand-specific polyadenylated RNA libraries" should also be made available as supplementary material.

There is only one time point sampled per species. I suggest discussing the possibility that the different pattern in *Anolis* might be caused partially by heterochrony in limb differentiation.

The authors present p-value distributions, which is very commendable, but I am surprised to see 10-40 genes called significant when the distribution is flat. I suggest checking results with the q-value of Storey (PNAS 100: 9440-9445; package in bioconductor), which is meant to interpret such distributions. The Benjamini-Hochberg FDR can call low p-values significant sometimes even when the distribution is flat because of the start of the iterations.

The Venn diagrams, as in Fig 3, are of course sensitive to cut-offs which are necessarily arbitrary. I suggest also checking correlation of fold-change between species, for genes which are called significant in at least one species, vs. a random set of genes of similar expression level.

Fig 4c and similar graphs: why show TPM rather than $\sqrt{\text{TPM}}$? This squashes low expression value changes.

Why perform PCA on TPM rather than $\sqrt{\text{TPM}}$, also?

The alligator embryos were from 6 nests, presumably outbred, but the mice from one inbred mother, and for the Anolis this is not specified. Please specify for Anolis, and discuss how this might impact the power to detect consistent expression changes.

Do I understand well that TPM was recomputed for each sub-set of genes? Please justify, as this is unusual and makes comparisons between figures more complicated.

The authors use median length for TPM computation. Please see the discussion of this strategy vs transcript aggregation in <https://www.biorxiv.org/content/early/2017/09/18/190199>

Please report all PCA bootstrap values as supplementary data.

Extended Data Table 1: what is the null expectation for the number of genes annotated to such phenotypes?

Minor remarks:

Discussion: "a high level of differential gene expression is evidence for distinct gene regulatory states". I agree, but I would suggest noting here that the evidence is even stronger because of the patterns of transcription factors.

Please use Celsius, not Fahrenheit, for temperatures.

Reviewer #3:

Remarks to the Author:
review of Stewart et al

The study intends to determine whether digit identity, meaning the A-P order in which digits exist on the autopod, has a distinctive gene expression profile. The authors use an impressive suite of methods to analyze the RNA transcriptomes from digit tissue in novel ways. It is a fascinating and complex set of data, and the result of evolutionary dynamism and differentiation is definitely interesting to evolutionary and developmental biologists who have sought an answer to these longstanding but elusive problems.

Methods:

Re: stage specific control - The authors sampled at a similar stage among species and in some species sampled multiple stages. All the selected stages were during the development of the phalanges, fairly late in digit development if one intends to find gene expression related to position. I suggest that comparing gene expression just prior to the developmental "decision" for how many digits to form, i.e. when the digit condensations are initiated in the metapodials, is arguably a better stage to use if one is interested in when positional identity is determined. Number of digits is determined in stage HH24-25 in the chick. This study as well as previous studies of transcriptomes of digit identity dissected digits at later stages, eg HH28. (Vargas and Fallon 2005; Wang, Young, Xue, and Wagner 2011 both use late stage tissue).

Digits vary in many ways, not only A-P position relative to the carpals/tarsals. Other features include number of phalanges, size and proportions of phalanges, size and proportion of digit lengths, order in which they arise developmentally. Therefore we might expect different genetic regulatory programs just as likely to differentiate these features as those differentiating position. In fact, looking at the later stages when these other features are being determined is more likely to pick up gene expression related to phenotypes emerging at this stage (e.g. relative growth and segmentation rates). Perhaps the authors could look at other features to see if they correlate better between the transcriptome data and late-stage digit morphogenesis.

Digit 1

The grouping of "known limb patterning genes" includes all genes sensitive to Shh signaling. The result of this analysis showed only mouse D1 differs from D2-3-4-5, and alligator D1-2-3 differs from D4-5.

Differentiating adjacent digits found 3 genes that differentiate D1 from D2, and one gene that differentiates D4 and D5 in all 3 species. None were found to distinguish the central 3 digits.

The re-analyzed chicken transcriptome data and human data also show D1 has a distinct profile compared with D2 and other digits.

From these results, the authors suggest a digit specific identity for D1. However, it could be equally concluded it is an "anterior-edge-specific" gene expression identity.

Furthermore, in a previous study, the chick forelimb is found to express precursors to five digits, with the D1 position arresting development after Sox9 expression (deBakker et al. 2013. Nature 500:445-448.)

Because of this contradictory evidence and the evolutionary dynamism shown in this study's data, I do not believe that these results are sufficient to conclude that the gene expression profile of the anteriormost digit in chicken wing is homologous to D1 when compared to the pentadactyl taxa.

General comment of the number of tests of the same data:

There seems to be a lot of reanalysis of transcriptomic data using many different statistical techniques. I am not a statistical expert but the number of repeated tests on the same data may have to be considered.

**Author response to all reviewers:**

We would like to thank the reviewers for their comments. We have revised the manuscript in response to
the issues raised and believe the study is significantly improved as a result. We first address two issues
raised by multiple reviewers before responding to other comments individually.

The reviewers correctly note that patterns of gene expression are dynamic in developing limbs.
Therefore, they asked for additional discussion of how our sampling strategy could impact the
conclusions of our study. We thank them for raising this issue and have modified the manuscript to
include such a discussion.

In this study, we consider two taxa for which limbs were sampled at multiple time points (chicken
at stages 28 and 31 and alligator at stages 18 and 19.5). These data allow for us to directly assess the
stability of expression profiles across a developmental window from (a) once digit condensations have
formed and interdigital mesenchyme has begun reducing to (b) after phalanges have formed but while
interdigital webbing remains. It is on the basis of these comparisons that we conclude our results are
robust, even though the limbs of mouse and *Anolis* were sampled at a single time point.

The developmental window that we study was selected on the basis of previous experimental
work. Dahn and Fallon (2000. *Science*, 289:438-41) demonstrated in the chicken hindlimb that genes
expressed in the interdigital mesenchyme pattern regulate digit-specific morphologies, including the
number of phalanges. Follow up work in the chicken hindlimb showed that this signaling occurs between
stages 27 and stage 30 (Suzuki *et al.* 2008. *PNAS*, 105:4185-4190). In our manuscript, we compare the
expression profiles chicken hindlimb digits of stages 28 and 31. Transcriptomic analysis shows that global
expression patterns of transcription factor genes (Extended Data Fig. 3 b), limb patterning genes
(Extended Data Fig. 3 c), and CDEGs (Fig. 5 c) are stable over this developmental window. This is also
observed in plots of individual genes of interest, which are stable between the two sampled time points
(Fig. 4 b, c; Fig. 6 b). Therefore, these data show that the signals pertinent to digit patterning are
expressed even at late stages of limb development, after phalanges have formed.

Analyses of alligator show a consistent pattern. Between stages 18 and 19.5, the global
expression patterns of limb patterning genes (Fig. 1 c) and CDEGs (Fig. 5 c) are stable across the
sampled developmental window. Additionally, plots of individual genes show a stable pattern between the
two stages sampled (Fig. 4 b, c; Fig. 6 c). Thus, the major conclusions presented in the study are unlikely
to be impacted by minor differences in staging between species sampled. Further evidence of this fact
can be seen in analyses that seek to identify correspondence between avian forewing digits and the digits
of pentadactyl species. These comparisons give the same results, regardless of whether we are
comparing avian forelimb digits sampled from stage 28 or 31 to the digits of other limbs (Fig 6 d). It is for
these reasons that we argue our results are robust, even though mouse and *Anolis* limbs were sampled
at a single time point within the developmental window described above.

We have modified the manuscript to include a more detailed explanation of why we sampled at
these time points (lines 275-290), and also how the data directly inform the question of sampling
robustness given the dynamism of gene expression in limb development (lines 130-132, Extended Data
Fig. 3 legend).

The reviewers also asked for additional supplementary materials to ensure reproducibility of our analyses.
We thank them for these suggestions and in helping us to make the study more fully transparent. We will
contribute all of the requested materials to the Dryad Digital Repository. Files include (1) gene lists (which
include unique gene IDs and gene names) of transcription factors, limb patterning genes, and 49 CDEGs
for all of the species we analyze (mouse, alligator, *Anolis*, chicken, and human); (2) the counts of mapped
reads from all of the newly sequenced data as well as the reanalyzed chicken data; (3) median transcript
lengths for all genes in all species; and (4) bootstrap support values for all PCA plots. We make note of
this in the revised *Data Availability* section. The files have not yet uploaded the files because, according
to the Dryad webpage, it is the policy of Nature Communications that files be uploaded after manuscript
acceptance.

Additionally, to further promote reproducibility, we have shared all R scripts that were used in this
study on GitHub (https://github.com/ThomasAStewart/digit_identity_project). From the code, one can
replicate all of the results and figures presented and in the manuscript starting from the mapped read files
(which will be available on Dryad). Although tidying and annotating these scripts for public usage required
significant effort, we believe it will make the study more accessible and useful to the community.

**Reviewer #1 Comments**

*This manuscript examines the notion of digit homology by testing whether the digits have*
*expression profiles that are conserved across amniotes. To this end the authors generate the*
*transcriptome of each digit separately in three amniote species (mouse, *Anolis* and alligator).*
*When considering a curated set of *Shh* responsive genes or the set of TFs, the HCA and PCA*
*analyses show clustering by digit except in *Anolis* where clustering is very weak suggesting*
*homogeneity across the autopod. These results are further confirmed by the analysis of the*
*distribution of p-values from different expression analysis. However, very few of the differentially*
*expressed genes between adjacent digits are conserved across species, also when considering*
*the chicken instead of *Anolis* except for the comparison between D1 and D2. These findings*
*support the already known unique profile of D1 across amniotes (here also shown for human) and*
*indicate that the expression profiles of the rest of the digits are species specific. This late*
*conclusion is novel and has strong significance for the field.*

Another important finding of this study is that digits at position 3 and 4 in the chick wing cluster

*with D3 and D4 in the other amniote limbs shown no evidence of transformation of their identities.*
*This leads the authors to suggest the alternative view that the digits in the bird wing are 1-3-4, as*
*previously considered by Xu and Mackem. I think that this proposal needs a deeper discussion*
*including its paleontology implications regarding the evolution of birds.*

We thank the referee for identifying this portion of the discussion as requiring additional detail. We
acknowledge that Xu and Mackem's and our model of digit identity leads to challenging questions with
respect to the interpretation of the fossil record, but neither of us has the detailed paleontological
knowledge to fully resolve this issue. The discussion of how the gene expression data presented here
relates to the paleontological record has been expanded, primarily emphasizing the need for more work
(lines 362-384).

*Overall I think that this is an interesting study with two main conclusions: i) the evolutionary*
*dynamism of the expression profiles of D2-D5 and ii) the interpretation that the digits in the bird*
*wing are 1-3-4. However, I think that several issues should be addressed before it can be*
*presented to the community.*

*A weakness of this study is that it is restricted to a single stage in mouse and in Anolis. It is well*
*known that the patterns of expression of many patterning genes, particularly those responding to*
*Shh signaling, are very dynamic and this dynamism may impact the discovery of conserved*
*expression patterns unless the stages between species and between samples are very well*
*matched. This is important regarding the conclusions of the "evolutionary dynamism" of digits. In*
*this regard it would be important to know the change in expression profiles between the two*
*alligator stages and also it would be desirable to analyze another stage in mouse. In any case the*
*temporal aspect should be widely discussed.*

We address this concern above, in the response to all reviewers. The manuscript has been edited to
include a discussion of how temporal dynamism in gene expression might impact the study.

*Along the same line, Hoxa13 is one of the 49 CDEGs identified. It has a dynamic pattern of*
*expression starting at the posterior-distal tip and expanding rapidly to cover the complete*
*autopod. It is considered the best marker for the autopod and clearly an important TF in the*
*formation of the digits but it is more difficult to understand a differential expression among the*
*digits unless this is fine/temporal differences in level of expression. The same applies to other*
*genes in the 49 CDEGs set.*

We, too, were surprised by the result that *Hoxa13* is differentially expressed across the autopod. As the
reviewer notes, the gene is typically described as a marker for the autopod, and its expression level is not
predicted to differ between the digits of a limb. Temporal dynamism is a possible explanation for this
pattern; however there might also be stable, fine-scale differences in expression level between the digits.
We find support for this latter interpretation from the literature: Woltering *et al.* (2014. PLOS Biology, 12
e1001773) present *in situ* hybridization data mouse E12.5 limb buds, and these images show
heterogeneity in staining intensity between digits. Robust staining is maintained across the distal
interdigital mesenchyme that spans between D1 and D2 (also between D4 and D5). By contrast, the
distal interdigital mesenchyme that spans between D2 and D3 (also between D3 and D4), shows a region
of weaker staining between the two digits. Our results allow for such reevaluation of published results
and, perhaps, reinterpretation of existing hypotheses of gene function. Regardless, as noted in the
'Response to all reviewers,' we have modified the manuscript to discuss how our sampling strategy and
temporal dynamics might impact the analyses and results of this study.

*Another point that raises some concern is the range of distribution of Pearson's correlation*
*coefficients, between 0.9-1 or even 0.98-1, two or three decimals appear to be needed to*
*appreciate differences.*

We have modified the figures that display Pearson's correlations so that differences between samples
can be more easily appreciated. Specifically, we have added black hash marks to denote minor intervals
between the major numbered intervals along the scale and in some cases text along the scale has been
modified to denote additional decimal places.

*Please indicate the significance of the bulk correction performed when the analyses involved*
*comparisons between developmental stages (Fig. 1, extended data Fig. 1 etc...) as well as the*
*meaning of the negative correlations.*

Bulk corrections were performed so that samples collected from the same limb at different stages or from
different types of limbs can be directly compared to one another. It is true that these corrections can affect
analyses (*e.g.*, the PCA plot of mean-shifted data can be different than that of non-corrected data).

However, we argue that this approach is justified in our analyses for two reasons: brevity, when results
presented by combining stages are the same as results of the stages being considered independently;
necessity, in order to compare the avian forelimb to other limbs.

When describing patterns of similarity among the digits of a limb that was sampled at multiple
stages, we present analyses of all samples simultaneously for brevity, to meet constraints in figure
number. For example, HCA and PCA of limb genes in alligator give similar results when run on (a) only
stage 18 samples, (b) only stage 19.5 samples, and (c) samples from both stages with bulk-correction.

Although the results of these analyses are not identical, we have written the manuscript to account for
differences between them. Specifically, samples of stage 18 clustering as (D1(D2, D3)(D4,D5), while
samples of stage 19.5 cluster as (D1,D2)D3)(D4,D5). Clustering of both stages with a bulk correction is
the same as 19.5 only. Thus, in the manuscript we have written, “*In alligator, an anterior cluster,*
*comprised of digits D1, D2, and D3, is differentiated from a posterior cluster, comprised of D4 and D5*”
(line 91). If samples of two stages are run simultaneously without a bulk correction, they show clustering
by stage (e.g., in PCA the samples are separated into two groups along PC1 or PC2 according to their
stage). Mean-shifting the data accounts for the effect of stage and allows the samples to group by digit
(e.g., Figs 1 c and 5 c; Extended Data Figs 3 and 6).

Similarly, when comparing the forelimb of chicken to other limbs, if no bulk correction is
performed, then samples cluster by limb. For example, when comparing chicken fore- and hindlimbs of
stage 28, the hindlimb samples usually form a single cluster to the exclusion of the forelimb samples;
similar patterns are observed when analyzing all genes, all transcription factors, limb patterning genes,
and CDEGs. And if samples of two limbs and of two stages are considered without bulk correction, then
the samples generally cluster first by stage and then by limb. By applying a bulk correction, samples of
the different limbs can group together, thereby allowing us to directly compare the digit transcriptomes
between limbs and, thus, to test hypotheses of homology.

The reviewer also asks that we explain the negative correlations of mean-shifted samples.
Negative correlation values arise because after bulk correction, a gene’s expression value is negative
when the sqrt(TPM) value is less than the mean sqrt(TPM) value of that gene among all samples of the
bulk. (Bulks being comprised of all samples from a particular stage or all samples of a particular limb.)

The above response, contains a more detailed explanation than what has been added to the
revised manuscript (lines 484-490). If the paper is accepted to *Nature Communications*, the authors will
opt to publish the peer review history of the paper, so that this response can be made available to
interested readers in full.

*Minor:*

*- The rationale to select this particular stage for the study, condensations formed and interdigital*
*spaces regressing, should be provided.*

This concern is addressed in the response to all reviewers and in the manuscript (lines 275-290).

*- I suggest changing the color used for D5 as, in some graphics, it is difficult to distinguish from*
*that used for D1.*

We thank the reviewer for this suggestion and have modified all figures accordingly.

- Because the curated set of Shh responsive genes, the list of 159 genes considered as “limb patterning genes”, has not been published (PhD source), I wonder whether it could be provided as extended data in this manuscript.

We thank the reviewer for this suggestion. The dissertation is currently online (open access), these data are already available to all researchers, and so it should be considered as published. Therefore, as noted in the response to all reviewers, we will make available on Dryad a table with this gene list and species-specific gene IDs.

- I am curious about how many of the 49 CDEGs are included in the set of the 159 limb patterning genes. Some of the genes included in the 49 CDEGs are

Of the 49 CDEGs, 13 are found in the 159 limb patterning genes. This information has been added included in the revised manuscript (line 202).

Reviewer #2 Comments

This is a very interesting study of digit transcriptome identity in 4 tetrapodes (plus human). The work and the interpretation are very careful, and this will be of interest to students of limb evolution, but more generally also for understanding the relation between anatomical homology and gene expression, a central question of Evo-Devo. I have a few comments on the methods and writing. It is especially important to make the study more reproducible. The authors lean on data and annotations from others, and must make theirs available for further studies.

Major comments:

As it is presented, the work is not reproducible. The authors need to make the following available:

- TPMs and counts as used for all samples;*
- gene list of "known patterning genes";*
- gene list of transcription factors as determined per species;*
- gene list for the 49 CDEGs.*

For all three gene lists, the authors should provide gene identifiers in all species, so that reproducibility of the work does not depend on future changes in orthology calls.

We thank Reviewer #2 for these suggestions for making the study more reproducible. As discussed in the
comments to all reviewers, we will make the following files available on Dryad: (1) a data table with all
gene lists and unique species identifiers, and (2) a table with mapped HTSeq counts for all samples with
calculated median gene lengths. We have provided count data and gene lengths rather than providing the
TPMs directly, because we believe that this data is more flexible. (TPMs can readily be calculated from
the information provided for any desired gene list.)

*The "in-house protocol (available upon request from Yale Genome Sequencing Center) used to*
*generate strand-specific polyadenylated RNA libraries" should also be made available as*
*supplementary material.*

Details of the protocol have been added to the methods section (lines 423-431).

*There is only one time point sampled per species. I suggest discussing the possibility that the*
*different pattern in Anolis might be caused partially by heterochrony in limb differentiation.*

This issue is addressed in the response to all reviewers, above.

*The authors present p-value distributions, which is very commendable, but I am surprised to see*
*10-40 genes called significant when the distribution is flat. I suggest checking results with the q-*
*value of Storey (PNAS 100: 9440-9445; package in bioconductor), which is meant to interpret*
*such distributions. The Benjamini-Hochberg FDR can call low p-values significant sometimes*
*even when the distribution is flat because of the start of the iterations.*

We thank the reviewer for this suggestion. One might note that 10-40 genes is a small number given the
thousands involved in a genome wide comparison and even a flat p value distribution can have that few
significant genes. We replicated all analyses using the q -value method of Storey. This method uniformly
calls more genes as significant at the FDR threshold of 0.05 than the Benjamini-Hochberg method
presented in the paper. Therefore, it is unlikely that the genes identified as differentially expressed by the
Benjamini-Hochberg FDR method are artifacts of the correction method.

Additionally, when the Storey method is applied, all of the major results of our study hold:
qualitative comparisons of differential expression across the autopod and between species are the same
(the discussion related to Figure 2), and we recover the same sets of genes as conserved and
differentially expressed between limbs (e.g., the central part of the Venn diagrams of Fig 3 and Fig 4 a
are identical, and the same 49 CDEGs are also recovered). We note this fact in the revised methods
section (lines 517-522). We do not include details of analyses related to the q -value of Storey in the
revised manuscript due to limited space and because these results can be replicated, if desired, from the

code and supplementary files that we will make available. (We provide code for both Benjamini-Hochberg
and Storey analyses.)

*The Venn diagrams, as in Fig 3, are of course sensitive to cut-offs which are necessarily arbitrary.*
*I suggest also checking correlation of fold-change between species, for genes which are called*
*significant in at least one species, vs. a random set of genes of similar expression level.*

We thank the reviewer for this suggestion. We agree that threshold stringency will change the number of
genes including in each subset, however, stricter thresholds will report a smaller number of genes in the
overlap set and, thus, result in higher p values from the binomial test. It is for this reason that we report
both the Venn diagrams and the binomial tests in the manuscript and regard our analyses as
conservative.

Nevertheless, we have implemented the suggested approach to assess the robustness of our
conclusions. Briefly, we considered genes differentially expressed at a given position for a given species
(e.g., between D1 and D2 in mouse) at a FDR of 0.05. Using this gene list, we calculated the correlation
of fold differences between the digits (e.g., D1 and D2) in the focal species and in other species (e.g.,
chicken and alligator). Then, we compared those correlation values to correlation values calculated for a
random gene set with similar expression levels. To summarize, we compared the distributions of
correlations between orthologs and random genes. The results of this new analysis mirrors results
recovered by the intersection of gene lists. We find strong evidence that the genes differentially
expressed between D1 and D2 behave consistently between species. By contrast, there is limited
evidence that the genes differentially expressed between more posterior digits behave differently than
randomly selected genes. These analyses are presented in the new Extended Data Figure 5 and
presented in the manuscript on lines 154-183 and 524-541.

*Fig 4c and similar graphs: why show TPM rather than sqrt(TPM)? This squashes low expression*
*value changes.*

We thank the reviewer for this suggestion, and we have modified all of these graphs accordingly.

*Why perform PCA on TPM rather than sqrt(TPM), also?*

Thank you for calling this to our attention. We did, in fact, perform PCA on sqrt(TPM) values. We have
made this clear in the revised manuscript (line 493).

*The alligator embryos were from 6 nests, presumably outbred, but the mice from one inbred*

*mother, and for the Anolis this is not specified. Please specify for Anolis, and discuss how this*
*might impact the power to detect consistent expression changes.*

We thank the reviewer for this suggestion and have modified the manuscript to include a discussion of
how factors like parentage and inbreeding might impact the power to detect differentially expressed
genes (lines 334-346).

*Do I understand well that TPM was recomputed for each sub-set of genes? Please justify, as this*
*is unusual and makes comparisons between figures more complicated.*

TPMs were calculated for each gene set in order to make between-species comparisons possible. (A
TPM value is relative to the total transcripts detected among the genes considered. Thus, the TPM value
of a given gene are only comparable between species if it is calculated with respect to the same gene
sets.) As the reviewer highlights, this comes at the cost of complicating comparisons between figures.

*The authors use median length for TPM computation. Please see the discussion of this strategy*
*vs transcript aggregation in <https://www.biorxiv.org/content/early/2017/09/18/190199>*

Thank you for highlighting this reference. We look forward using this new method in future studies. In the
current short read data we have, transcript levels are not directly quantifiable and could be only estimated
based on model assumptions that we cannot directly assess.

*Please report all PCA bootstrap values as supplementary data.*

These values have been calculated for all plots and will be made available as files uploaded to Dryad
upon the manuscript's acceptance. This is now noted in the revised *Data Availability* section.

*Extended Data Table 1: what is the null expectation for the number of genes annotated to such*
*phenotypes?*

We thank the reviewer for raising this question. Unfortunately, we are unable to provide a simple answer,
because of both pragmatic and principled reasons. First, it is difficult to extract this information from some
of the online databases used to compile the table. For example, using the Human Gene Mutation
Database, searching for disease phenotypes with the word "limb" returns only 10 results and does not
return the many named syndromes (*e.g.*, Werner mesomelic syndrome) with limb phenotypes. In our
study, we queried this database manually, searching gene-by-gene to identify named syndromes and
then tracking down whether the syndromes have limb phenotypes. Thus, it is challenging for us to know

the number of genes with limb-associated phenotypes we might expect by chance if we were to randomly
sample 49 genes across the databases.

Second, these phenotype data bases are inherently non-random, because they are influenced by
the relative success of different research programs, with successful research programs representing a
higher number of entries than research on genes from less well funded labs. We therefore regard
Extended Data Table 1 as descriptive rather than statistical (*i.e.*, we do not attach a *p* value to the fraction
of genes with known limb phenotypes).

*Minor remarks:*

*Discussion: "a high level of differential gene expression is evidence for distinct gene regulatory*
*states". I agree, but I would suggest noting here that the evidence is even stronger because of*
*the patterns of transcription factors.*

We thank the reviewer for this suggestion and have modified the text accordingly (lines 273-274)

*Please use Celsius, not Farenheit, for temperatures.*

Thank you - this has been corrected.

**Reviewer #3 Comments**

*review of Stewart et al*

*The study intends to determine whether digit identity, meaning the A-P order in which digits exist*
*on the autopod, has a distinctive gene expression profile. The authors use an impressive suite of*
*methods to analyze the RNA transcriptomes from digit tissue in novel ways. It is a fascinating and*
*complex set of data, and the result of evolutionary dynamism and differentiation is definitely*
*interesting to evolutionary and developmental biologists who have sought an answer to these*
*longstanding but elusive problems.*

The first sentence of this review may reveal a misunderstanding, or disagreement, between the reviewer
and the authors. In the manuscript, we explicitly do not equate digit identity with AP position. On lines 35-
37 we state: "*it remains controversial whether such hypotheses of identity correspond to distinct*
*developmental programs among the digits (developmental identities), or just the relative position of digits*
*along the limb's anteroposterior axis (positional identities)."*

We do not make the assumption that digit position and developmental identity are the same thing,
because homeotic mutations show that the location of a body part and its developmental identity are not
necessarily associated with each other. Reviewer #3's use of position to define digit identity has
implications for other comments, including the one that follows immediately below.

*Methods:*

*Re: stage specific control - The authors sampled at a similar stage among species and in some*
*species sampled multiple stages. All the selected stages were during the development of the*
*phalanges, fairly late in digit development if one intends to find gene expression related to*
*position. I suggest that comparing gene expression just prior to the developmental "decision" for*
*how many digits to form, i.e. when the digit condensations are initiated in the metapodials, is*
*arguably a better stage to use if one is interested in when positional identity is determined.*
*Number of digits is determined in stage HH24-25 in the chick. This study as well as previous*
*studies of transcriptomes of digit identity dissected digits at later stages, eg HH28. (Vargas and*
*Fallon 2005; Wang, Young, Xue, and Wagner 2011 both use late stage tissue).*

We agree with the Reviewer that experimental evidence shows *Shh* signaling determines the number of
digits eventually formed and also initiates the development of differences between digits. However, the
execution of those programs and, thus, the proximate mechanisms of digit identity are executed long after
the *Shh* signal has ceased. Furthermore, it has been shown that "Shh target genes" are maintaining their
expression long after the Shh signal stopped due to protein-protein interactions between the Gi3 and the
Hox proteins (Chen et al., 2004, Development 131:2339-47). As discussed above, in the response to all
reviewers, it is signaling in the interdigital mesenchyme that is directly responsible for regulating digit-
specific morphologies once digit condensations have started to form.

*Digits vary in many ways, not only A-P position relative to the carpals/tarsals. Other features*
*include number of phalanges, size and proportions of phalanges, size and proportion of digit*
*lengths, order in which they arise developmentally. Therefore we might expect different genetic*
*regulatory programs just as likely to differentiate these features as those differentiating position.*
*In fact, looking at the later stages when these other features are being determined is more likely*
*to pick up gene expression related to phenotypes emerging at this stage (e.g. relative growth and*
*segmentation rates). Perhaps the authors could look at other features to see if they correlate*
*better between the transcriptome data and late-stage digit morphogenesis.*

We thank the reviewer for this suggestion. Indeed, linking description of expression profiles to
morphological signatures, like phalangeal number or the presence and absence of claws, is an extremely
useful application of these data, but beyond the scope of this manuscript.

Digit 1

The grouping of “known limb patterning genes” includes all genes sensitive to Shh signaling. The result of this analysis showed only mouse D1 differs from D2-3-4-5, and alligator D1-2-3 differs from D4-5.

Differentiating adjacent digits found 3 genes that differentiate D1 from D2, and one gene that differentiates D4 and D5 in all 3 species. None were found to distinguish the central 3 digits.

The re-analyzed chicken transcriptome data and human data also show D1 has a distinct profile compared with D2 and other digits.

From these results, the authors suggest a digit specific identity for D1. However, it could be equally concluded it is an “anterior-edge-specific” gene expression identity.

We thank the Reviewer for raising the issue of edge effects, and we have modified the manuscript to clarify this issue. Specifically, the Reviewer raises the possibility that the unique gene-expression profile of D1 in pentadactyl limbs should be understood as an ‘anterior-edge-specific’ gene expression profile. However, we argue that previously published studies do not support this hypothesis. The hindlimb of rabbit has lost its anterior digit. In development, this limb has also lost the characteristic gene expression profile of D1; there is no territory of reduced *HoxD10* and *HoxD12* expression (Salinas-Saavedra et al. 2014. *Front Zool.* 2014; 11: 33). Additionally, a study of Silkie chicken mutants, which have additional anterior digit on their foot, showed that the digit with the morphology of D1 (which is now no longer the most anterior digit) still has gene expression pattern of the native digit 1 (Vargas and Fallon. 2005, *J. Exp. Zool. (Mol. Dev. Evol.)* 304B:86-90). Thus, we do not think that we are dealing with an edge effect, but rather a genuine reflection of the developmental identity of digit 1. This is discussed in the revised manuscript (lines 295-301).

*Furthermore, in a previous study, the chick forelimb is found to express precursors to five digits, with the D1 position arresting development after Sox9 expression (deBakker et al. 2013. *Nature* 500:445-448.)*

Because of this contradictory evidence and the evolutionary dynamism shown in this study's data, I do not believe that these results are sufficient to conclude that the gene expression profile of the anteriormost digit in chicken wing is homologous to D1 when compared to the pentadactyl taxa.

We agree with the results of deBakker et al. (2013) and point out that this conclusion was reached
previously by four different labs using four different methods. Pre-cartilaginous digit condensations were
described in the chicken forelimb bud by Larsson and Wagner (2002. J. Exp. Zool. (Mol. Dev. Evol.)
294:146-151), Kundrát et al. (2002. J. Exp. Zool. (Mol. Dev. Evol.) 294:152-159) and van Welten et al.
(2005. Evol. Dev. 7:18-28); and five cartilages were described in the ostrich forelimb bud by Feduccia and
Nowicki (2002. Sci. Nat. 89:391-393). Indeed, we were among the first to propose that the chicken wing
bud has five such condensations. So, we concur that five digit progenitors develop early in the chicken
wing and that the digits in the adult wing develop from positions D2, D3, and D4. We make the point
repeatedly throughout the manuscript. However, this fact is not contradictory with data presented in the
manuscript, because digit position and developmental identity are dissociable, as is the case with other
body parts, a phenomenon called homeosis.

The reviewer's primary concern here appears to be that insufficient evidence has been provided
to support diagnosis of similarity in the gene expression profiles of digit D2 of the bird wing and the digit
D1 of other limbs. However, we address this explicitly in our study. Our analyses confirm the previous
results of Wang et al (2011. Nature, 477:583–586) and other publications (e.g., Vargas and Fallon. 2005.
453 J. Exp. Zool. (Mol. Dev. Evol.) 304B:86-90). To reiterate, we show that the digit in the second position of
454 the wing (D2) corresponds with respect to gene expression profile to the digit D1 of the chicken hindlimb
through analyses of several gene sets (e.g., Fig 6 a; Extended Data Fig. 6 b, c) and inspection of
individual genes (Fig. 6 b). We additionally describe correspondence between D2 of the wing and D1 of
pentadactyl limbs (Fig. 6 d).

*General comment of the number of tests of the same data:*

*There seems to be a lot of reanalysis of transcriptomic data using many different statistical*
*techniques. I am not a statistical expert but the number of repeated tests on the same data may*
*have to be considered.*

When many statistical analyses are run on a single data set (as for discovering differential expression of
genes between two kinds of samples), we have corrected for these multiple tests. (We adjusted p values
by the Benjamini Hochberg procedure and also considered the q -value of Storey). However, when
different aspects of the data structures are assessed with different methods the purpose is to test for the
robustness of these results with respect to statistical method. For the latter purpose no addition
corrections are necessary.

Reviewers' Comments:

Reviewer #1:

None

Reviewer #2:

Remarks to the Author:

I thank the authors for their thorough revision and reply to reviews. I have no further major comments, and I think that this is a very interesting study.

One very minor comment: what does "available upon request" mean in

"All code used for analyses is available on GitHub

(https://github.com/ThomasASTewart/digit_identity_project) and available upon request"

Reviewer #3:

Remarks to the Author:

I thank the authors for their rapid responses to my comments.

I have read this manuscript through numerous times and still cannot resolve the aim of the study with the results and interpretation of the study. I will try to articulate my continuing difficulty with the manuscript and see if it can be addressed.

The authors dispute my characterizing their definition of digit identity as A-P order (first paragraph of reviewer 3 response). Since the aim of the study is to homologize digits of different taxa based on transcriptional profile, then my first question is whether they are trying to homologize positionally-defined digits (A-P order) or morphologically-defined digits (number of phalanges, length, etc). Is there another option? Yes, a third category is positionally-defined digit identity after frame-shifted or otherwise tinkered A-P order (through digit reduction), which could happen shortly after digit metapodial initiation but certainly before any phalanges form.

If they are homologizing by positional identity, then they should aim for gene expression related to A-P position, which is why the late stage of their samples is potentially problematic, because A-P order is already set.

But then, In the response-to-reviewers regarding the stages they studied, the authors write :

"The developmental window that we study was selected on the basis of previous experimental work. Dahn and Fallon (2000. Science, 289:438-41) demonstrated in the chicken hindlimb that genes expressed in the interdigital mesenchyme pattern regulate digit-specific morphologies, including the number of phalanges"

This comment led me to think that they are referring to morphologically-defined digit identity, rather than positionally defined digit identity.

Again they write in line 386 of the response to reviewers' concern about appropriate stages "As discussed above, in the response to all reviewers, it is signaling in the interdigital mesenchyme that is directly responsible for regulating digit- specific morphologies once digit condensations have started to form. "

But then just below that they state that "linking description of expression profiles to morphological

signatures" are "beyond the scope of this manuscript". ??? I hope you can understand my confusion! If they *are* trying to homologize by morphology, then in their analysis, they don't actually explicitly compare morphologically-similar digits, which leads me back to my original thought that they are comparing by A-P position.

Lastly, in the summary, they state "Here we show dramatic evolutionary dynamism in the gene expression profiles of digits, challenging the notion that five digit identities are conserved across amniotes".

If they mean morphologically-defined digit identity, it's obvious they aren't conserved, because morphology - numbers of phalanges, digit lengths, widths, claws, etc. - varies a lot among taxa, and there would be no expectation of different morphologies among digits being regulated by similar gene expression. If they mean positionally-defined digit identities, it's obvious digit identities are conserved, for about 300 million years, simply because they exist as pentadactyl (at least initially in development). Their results of "evolutionary dynamism" suggests the developmental gene regulation has evolved even while the five digits have stabilized.

And therefore, using the diverged or drifted gene regulation of other taxa to hypothesize about bird wing digit identity seems weak support. [Last sentence of the summary: ". . the identity of the anterior-most digit has shifted position, suggestion a 1,3,4 digit identity in the bird wing".]

**Reviewers' comments:**

**Reviewer #2 (Remarks to the Author):**

*I thank the authors for their thorough revision and reply to reviews. I have no further major*
*comments, and I think that this is a very interesting study.*

*One very minor comment: what does "available upon request" mean in*

*"All code used for analyses is available on GitHub*

*(https://github.com/ThomasAStewart/digit_identity_project) and available upon request"*

Thank you for calling attention to the phrasing of this statement. We have deleted the "available upon
request" phrase; it was only intended to convey that code also will be made available by email, if
requested.

**Reviewer #3 (Remarks to the Author):**

*I thank the authors for their rapid responses to my comments.*

*I have read this manuscript through numerous times and still cannot resolve the aim of the study*
*with the results and interpretation of the study. I will try to articulate my continuing difficulty with*
*the manuscript and see if it can be addressed.*

We sincerely thank the reviewer for explaining their difficulties in resolving the aim of our study. We hope
that this confusion can be clarified through our responses to their questions and the edits made to the
manuscript to more fully explain our experimental approach.

*The authors dispute my characterizing their definition of digit identity as A-P order (first paragraph*
*of reviewer 3 response). Since the aim of the study is to homologize digits of different taxa based*
*on transcriptional profile, then my first question is whether they are trying to homologize*
*positionally-defined digits (A-P order) or morphologically-defined digits (number of phalanges,*
*length, etc). Is there another option? Yes, a third category is positionally-defined digit identity after*
*frame-shifted or otherwise tinkered A-P order (through digit reduction), which could happen*
*shortly after digit metapodial initiation but certainly before any phalanges form.*

To be clear: the aim of this study is not to analyze the morphologies of individual digits (e.g., phalangeal
number or length), and we are not homologizing digits according to their position. Yes, we think that there

is another strategy of homologizing digits, albeit different from the one the reviewer proposes. Namely, we
think that digit identity is grounded in the genetic control of digit morphogenesis. Based on this idea, digit
identity and homology can be assessed on the basis of developmental gene expression profiles
(transcription factors, signaling molecules, *etc.*). This approach was pioneered in our 2011 paper in
*Nature*, and it is routinely used in the assessment of cell type homology in other contexts. Specifically, the
idea is that homologous digits are expected to express the same or similar sets of developmental genes,
and non-homologous digits express substantially different sets of developmental genes. Here, rather than
assuming that this applies to all digits in amniotes, this paper tests whether all digits can be distinguished
by their developmental gene expression profiles.

Our position is motivated by two ideas: (1) *Owen's original and broadly accepted definition that*
*homologs can have different shapes*. Richard Owen's 1848 definition of homology reads "the same
character in different animals regardless of form and function." An important implication of Owen's
definition is that although similarity can provide evidence in favor of homology, it does not "define" it.
There are many well-known cases where homologs do not have similar shapes. For example, homology
of the primary jaw articulation of reptiles (and other non-mammalian gnathostomes) and two of the inner
ear ossicles of mammals (the incus and malleus) are not similar in function, shape or position. The
homology of these elements was established through a series of transformation stages in the fossil
record. This example shows how "the same" character (*i.e.*, a character traceable through phylogeny) can
have dramatically different shapes and functions in derived species. (2) *The idea that the "sameness" in*
*Owen's definition refers to a developmental fact, namely that during development the activation of a gene*
*regulatory network will individuate (i.e., differentiate) a character from the rest of the body*. This second
idea has been proposed by a number of researchers independently. For example, these gene regulatory
networks have been called "kernels" by Davidson and Erwin, "selector modules" by Oliver Hobert, and
"character identity networks" by one of our authors, GPW (described in a 2007 *Nature Reviews Genetics*
paper and elaborated in his 2014 book "Homology, Genes and Evolutionary Innovation"). It has further
been recognized that these, let us call them "core networks," tend to be rather conservative across
species.

In this paper, we test the hypothesis that the five digits of the pentadactyl amniote limb have
distinct "developmental identities" (*i.e.*, each digit has its own "core network") and that these are
conserved across all major amniote clades. A prediction of this hypothesis is that there should be
consistent patterns of transcription factor gene expression to be found in corresponding digits across
these lineages. The rationale is that it has been shown in many cases that the developmental identity of a
character is established through the activation of a small network of transcription factor genes (*e.g.*,
Pax/Six/Eya network for eye development). We find that our expression data rejects this prediction of a
conserved developmental identity for each digit in the amniote pentadactyl limb, the exception is the digit
that occupies the anterior-most position (*i.e.*, the "thumb").

*If they are homologizing by positional identity, then they should aim for gene expression related to*
*A-P position, which is why the late stage of their samples is potentially problematic, because A-P*
*order is already set.*

*But then, In the response-to-reviewers regarding the stages they studied, the authors write :*

*“The developmental window that we study was selected on the basis of previous experimental*
*work. Dahn and Fallon (2000. Science, 289:438-41) demonstrated in the chicken hindlimb that*
*genes expressed in the interdigital mesenchyme pattern regulate digit-specific morphologies,*
*including the number of phalanges”*

*This comment led me to think that they are referring to morphologically-defined digit identity,*
*rather than positionally defined digit identity.*

While it is true that we are analyzing gene expression at a time when features of digits (e.g., number of
phalanges) are being established, our aim was to document and compare the expression of regulatory
genes that govern the morphogenesis of digits. The developmental stages targeted in our study are
specifically chosen to capture the stage of development where the digit identity is realized, and, thus,
when the genes responsible for controlling this process are expressed. These stages of development
have been identified in previous experimental work by Dahn and Fallon (cited in the manuscript as #13).
We have modified the introduction to make this point clearer (lines 69-79).

*Again they write in line 386 of the response to reviewers' concern about appropriate stages “As*
*discussed above, in the response to all reviewers, it is signaling in the interdigital mesenchyme*
*that is directly responsible for regulating digit- specific morphologies once digit condensations*
*have started to form. “*

*But then just below that they state that “linking description of expression profiles to morphological*
*signatures” are “beyond the scope of this manuscript”. ??? I hope you can understand my*
*confusion! If they *are* trying to homologize by morphology, then in their analysis, they don't*
*actually explicitly compare morphologically-similar digits, which leads me back to my original*
*thought that they are comparing by A-P position.*

We thank the reviewer for these comments; they helped for us to understand a source of
misunderstanding. Here the reviewer expresses confusion as to why the time point we selected, after
digits have started to form, could meaningfully inform the question of digit homology. In the previous
submission, the discussion section included citations that explained why we selected this timepoint was
selected. To make our reasoning clearer, we have revised the introduction and discuss the time point we
studied and why we studied gene expression after digit condensation (lines 69-79). We also more
explicitly discuss how our data related to gene expression studies that have considered earlier stages
(lines 267-275).

In the previous round of reviews, we agreed with the reviewer that our data will be useful for
projects that aim to diagnose the genetic basis of various digit-specific phenotypes (e.g., the presence or
absence of claws). However, we maintain that such analyses are beyond the scope of this study, which
focuses on digit homology.

*Lastly, in the summary, they state “Here we show dramatic evolutionary dynamism in the gene*
*expression profiles of digits, challenging the notion that five digit identities are conserved across*
*amniotes”.*

*If they mean morphologically-defined digit identity, it’s obvious they aren’t conserved, because*
*morphology - numbers of phalanges, digit lengths, widths, claws, etc. - varies a lot among taxa,*
*and there would be no expectation of different morphologies among digits being regulated by*
*similar gene expression. If they mean positionally-defined digit identities, it’s obvious digit*
*identities are conserved, for about 300 million years, simply because they exist as pentadactyl (at*
*least initially in development). Their results of “evolutionary dynamism” suggests the*
*developmental gene regulation has evolved even while the five digits have stabilized.*

In the final sentence of this paragraph, the reviewer articulates one of the major findings of our study. We
are glad that this point was clear. It is, we believe, a significant result that challenges many assumptions
that motivate contemporary discussions about the homology of digits, and it has implications for
hypotheses about the evolutionary origin of the bird wing, discussed below.

*And therefore, using the diverged or drifted gene regulation of other taxa to hypothesize about*
*bird wing digit identity seems weak support. [Last sentence of the summary: “.. the identity of the*
*anterior-most digit has shifted position, suggestion a 1,3,4 digit identity in the bird wing”.]*

The reviewer rightly notes that it is challenging to diagnose identity of bird wing digits by comparison to
the limbs of other species. We acknowledge this explicitly in our writing (lines 263-266). However, our
conclusion that the posterior two digits of bird wing have not undergone a homeotic shift is not based

solely upon comparison to pentadactyl limbs; it is also based upon a series of comparisons with the avian
hindlimb, and also independently supported based on different evidence by Xu and Mackem (2013, cited
in the manuscript as #41).

*Stepping back from the details, it seems that the primary source of reviewer #3's confusion with this study*
*stems from the fact that they did not recognize our position that the identity (=homology) of morphological*
*characters can be rooted in the activation of a conserved gene regulatory network. If this idea is left out*
*from consideration, our paper will not make sense to them. Nevertheless, the accumulated evidence from*
*developmental biology from the last three decades supports the notion that character identity*
*(=homology) is rooted in the activity of shared gene regulatory networks responsible for the*
*developmental individuation of morphological characters. Describing exactly which part of the*
*developmental cascade represents the core network (clearly not all parts of development are conserved*
*among homologous characters) goes beyond the scope of this brief statement, but can be found in the*
*published literature.*